# Twisted-bilayer FeSe and the Fe-based superlattices

**Paul M. Eugenio$^{\star\dagger}$ and Oskar Vafek**

National High Magnetic Field Laboratory, Tallahassee, Florida, 32304, USA
Department of Physics, Florida State University, Tallahassee, Florida 32306, USA

$\star$ eugenio@magnet.fsu.edu , $\dagger$ paul.eugenio@uconn.edu

## Abstract

We derive BM-like continuum models for the bands of superlattice heterostructures formed out of Fe-chalcogenide monolayers: (I) a single monolayer experiencing an external periodic potential, and (II) twisted bilayers with long-range moire tunneling. A symmetry derivation for the inter-layer moire tunnelling is provided for both the $\Gamma$ and $M$ high-symmetry points. In this paper, we focus on moire bands formed from hole-band maxima centered on $\Gamma$, and show the possibility of moire bands with $C = 0$ or $\pm 1$ topological quantum numbers without breaking time-reversal symmetry. In the $C = 0$ region for $\theta \to 0$ (and similarly in the limit of large superlattice period for I), the system becomes a square lattice of 2D harmonic oscillators. We fit our model to FeSe and argue that it is a viable platform for the simulation of the square Hubbard model with tunable interaction strength.



# 1   Introduction

The 2018 discovery of nearly flat bands in twisted bilayer graphene (tBG) marked the genesis of a new era of highly tunable devices which combine strong coupling physics with non-trivial topology [1,2]. Characteristic of such devices is the existence of a superlattice potential with a period orders of magnitude larger than the atomic scale, which drives long-wavelength inter-layer tunneling, and produces a mini Brillouin zone (mBZ) orders of magnitude smaller than the BZ of the original graphene monolayer [3–11].

In tBG and other moire heterostructures, the superlattice forms as an emergent moire pattern from the overlapping crystal bilayers, giving rise to bands whose bandwidths are controlled by the angle of the twist [3,4]. For an appropriately chosen twist angle, the bandwidth of the lowest energy bands shrinks to the order of the interaction energy, effectively engineering strong interactions [4].

This stack-and-twist approach to creating moire superlattices has been used to engineer flatbands in other materials, including the transition metal dichalcogenides [12–21] and chirally-stacked graphenes atop hexaboron nitride [22,23], to name a few. Additionally, superlattice potentials have been engineered in monolayers through spatially periodic dielectric screening (SPDS), where the Coulomb potential is spatially modulated via a dielectric substrate, producing a mBZ for the renormalized bands [24,25].

Here we propose a new class of superlattice materials composed of monolayer Fe-chalcogenides [26–65]. Such monolayers are interesting on their own, combining multi-band physics with spin-orbit coupling to produce a variety of phenomena [59–63, 66–69], such as unconventional high-temperature superconductivity in monolayer FeSe upon doping, with $T_c$'s reported as high as 65 K [26] and even 109 K [27].

Unlike graphene however, the low-energy properties of the iron-chalcogenides cannot be accurately captured from a tight-binding model with less than 5 bands [70,71]. This picture is further complicated by the moderate renormalization of the bands due to interactions, which is stronger than what can be predicted within the assumptions of ab initio methods like density functional theory (DFT) [72–80], thus placing a barrier on the accurate determination of microscopic parameters.

Despite these difficulties, an accurate minimal low-energy model for the modes near the Fermi level can still be constructed on the basis of symmetry. This is because the iron-chalcogenides are known for having low-energy bands which disperse sharply in the vicinity of high symmetry points – two hole-band maxima at $\Gamma$, and four electronic-like bands at $M$ – and in the case where those bands cross the Fermi level, produce Fermi surfaces which are small relative to the atomic Brillouin zone [59]. Therefore, those electronic states which are most relevant to the low-energy physics can be described within a $\mathbf{k} \cdot \mathbf{p}$ effective theory for the fields at $\mathcal{Q}$, where $\mathcal{Q} \in \{\Gamma, M\}$.

From a microscopic perspective, these slowly varying effective fields, which we write as $\psi_{\mathcal{Q},a}(\mathbf{x})$, are the low energy contribution to $d_a(\mathbf{x})$, the annihilation operator for the atomic

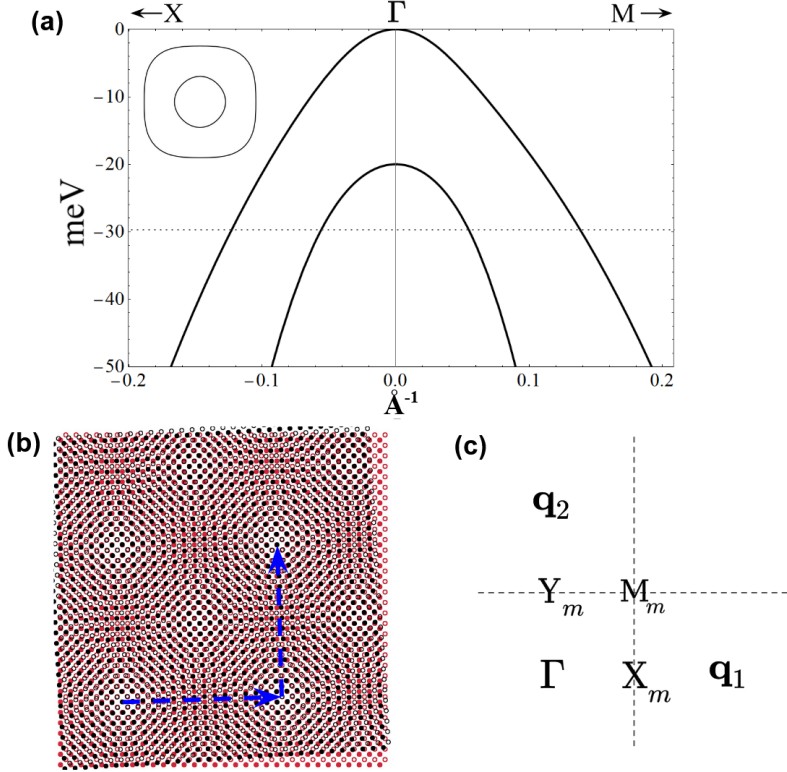

Figure 1: (a) The continuum bands about Γ, fit to photoemission data for FeSe [58]. The inset represents an example of the energy contours at a non-specific energy indicated by the dashed line. (b) Twisted Fe-Fe planes. The chalcogenide atoms have been suppressed in this drawing for clarity, and instead non-equivalent Fe's within the same unit cell are drawn as closed/open circles. The dashed blue lines connect equivalent moire lattice sites. (c) The superlattice Brillouin zone centered on its corner $M_m = (q_S/2, q_S/2)$, and showing the neighboring zones at $\mathbf{q}_1 = (q_S, 0)$ and $\mathbf{q}_2 = (0, q_S)$.

orbital $a$ at discrete lattice site $\mathbf{x}$:

$$d_a(\mathbf{x}) = \sum_{\mathcal{Q}} e^{i\mathcal{Q}\cdot\mathbf{x}} \psi_{\mathcal{Q},a}(\mathbf{x}) + \text{higher energy states.} \tag{1}$$

However, no microscopic information is actually needed to derive the general form of the effective theory, only *how* the relevant fields at $\mathcal{Q}$ transform under space group symmetries – of which they are guaranteed to be eigenstates because $\mathcal{Q}$ is a high symmetry point. For instance, noting $\alpha \in \{\uparrow, \downarrow\}$, the effective theory for the relevant fields at Γ,

$$h_\Gamma^{\alpha\beta}(-i\boldsymbol{\nabla}) = \begin{pmatrix} \epsilon_\Gamma - \mu\boldsymbol{\nabla}^2 - a\partial_x\partial_y & -b(\partial_x^2 - \partial_y^2) \\ -b(\partial_x^2 - \partial_y^2) & \epsilon_\Gamma - \mu\boldsymbol{\nabla}^2 + a\partial_x\partial_y \end{pmatrix} \delta^{\alpha\beta} + \lambda \begin{pmatrix} 0 & -i \\ i & 0 \end{pmatrix} s_z^{\alpha\beta}, \tag{2}$$

was derived previously by Cvetkovic and one of us [70], using only the fact that those fields transform as the doublet $\psi_\Gamma \sim (d_{Yz}, -d_{Xz})$, which is an irreducible representation of the space group at Γ. We fit its invariants directly to photoemission data in bulk FeSe above its structural transition [58] (see Fig 1-a): $(\mu, a, b) = (-2830, -3440.93, -a/2.5)$ meVÅ$^2$ and $\lambda = 10$ meV. The fitted theory fully accounts for the renormalization due to the interactions at sufficiently long wavelengths and accurately describes the observed bands.

Therefore, we work within the effective theories for the fields about $\Gamma$ and $M$, and propose two classes of materials: (I) a single FeSe monolayer experiencing an external periodic potential (say via SPDS); and (II) uniformly-rotated small-angle twisted bilayers of FeSe, which experience a moire inter-layer tunneling. In both cases, we derive a continuum description for the superlattice potential (or tunneling) using the irreducible representations (irreps) of the space group at $\Gamma$ and $M$ [70, 81, 82] – the details for which can be found in the following sections. We work within the assumption that the superlattice potential is dominated by scatterings with the smallest wavevectors; are spin independent; and, in the case of (II), leading order in the twist angle $\theta$.

In particular, we focus on chalcogenide monolayers with hole-band maxima centered at $\Gamma$, such as is true for thin films of FeSe [28, 29, 36, 49] as well as the underdoped variant of monolayer FeSe/SrTiO$_3$ [28], where a hole band maximum lies 10 meV below the Fermi level. We generally refer to "hole-band maxima" (plural) as opposed to "maximum" (singular) because the upper hole band is expected to arise from a quadratic-band touching described by Eqn 2, but one where the presence of $\mu$ forces both bands to disperse downward. These bands are then gapped due to the spin-orbit $\lambda$ ($= 10$ meV reported in the bulk [59]).

The effective Hamiltonian for case (II) about $\Gamma$ describes two relatively rotated and coupled copies of Eqn 2,

$$
\begin{aligned}
H_{\text{moire},\Gamma} = \int d^2\mathbf{x}\; & \psi^\dagger_{\Gamma,1,\alpha}(\mathbf{x}) h^{\alpha\beta}_\Gamma(-i\boldsymbol{\nabla})\psi_{\Gamma,1,\beta}(\mathbf{x}) + \psi^\dagger_{\Gamma,2,\alpha}(\mathbf{x}) h^{\alpha\beta}_\Gamma(-i\mathcal{R}^{-1}_\theta\boldsymbol{\nabla})\psi_{\Gamma,2,\beta}(\mathbf{x}) \\
& + 2(w_{0,\text{os}}\delta^{ll'} + w_{0,\text{t}}\sigma^{ll'}_1)\,\psi^\dagger_{\Gamma,l,\alpha}(\mathbf{x})\begin{pmatrix}1 & 0 \\ 0 & 1\end{pmatrix}\Big[\cos(\mathbf{q}_1\cdot\mathbf{x}) + \cos(\mathbf{q}_2\cdot\mathbf{x})\Big]\psi_{\Gamma,l',\alpha}(\mathbf{x}) \\
& + 2(w_{1,\text{os}}\delta^{ll'} + w_{1,\text{t}}\sigma^{ll'}_1)\,\psi^\dagger_{\Gamma,l,\alpha}(\mathbf{x})\begin{pmatrix}0 & 1 \\ 1 & 0\end{pmatrix}\Big[\cos(\mathbf{q}_1\cdot\mathbf{x}) - \cos(\mathbf{q}_2\cdot\mathbf{x})\Big]\psi_{\Gamma,l',\alpha}(\mathbf{x}) \\
& + t\,\psi^\dagger_{\Gamma,1,\alpha}(\mathbf{x})\begin{pmatrix}1 & 0 \\ 0 & 1\end{pmatrix}\psi_{\Gamma,2,\alpha}(\mathbf{x}) + \text{h.c.}
\end{aligned}
\tag{3}
$$

The invariants $w_{0,\text{os}}, w_{1,\text{os}}$ reflect the long-wavelength component of the symmetry-allowed variations in the on-site energy due to the presence of the moire pattern [12, 13, 17]. The form of the tunneling invariants $t, w_{0,\text{t}}, w_{1,\text{t}}$ follows consistently from both a microscopic calculation [83, 84], as well as from the leading order part of the effective tunneling $T(\mathbf{u})$ within the elasticity theory for displaced bilayers [5], where $\mathbf{u}(\mathbf{x}) \equiv \mathbf{r}_1(\mathbf{x}) - \mathbf{r}_2(\mathbf{x})$ is the displacement between the two layers $l \in \{1, 2\}$ as a function of the lab coordinate $\mathbf{x}$. (WLOG, we choose our lab coordinates to be the unrotated frame of one of the layers $\mathbf{r}_1 = \mathbf{x}$.) The $(\mathbf{q}_a)_j = q_S\delta_{a,j}$ are the reciprocal lattice vectors, whose size $q_S = 2\pi/l_S$ is set by the superlattice period $l_S = l_{\text{Fe}}/\theta$, where $l_{\text{Fe}}$ is size of the Fe unit cell.

Notice that the intra-layer Hamiltonian only depends on the twist angle through $h^{\alpha\beta}_\Gamma(-i\mathcal{R}^{-1}_\theta\boldsymbol{\nabla})$, and that no term exists which relatively shifts the quadratic band touchings between different layers in $\mathbf{k}$-space. This is unlike the case for the fields about a non-zero crystal momentum – such as for the $K$ point in graphene or the $M$ point in FeSe – and is a consequence of the fact that the twist in $\mathbf{k}$-space is taken around $\Gamma$. The remaining $h_\Gamma(-i\mathcal{R}^{-1}_\theta\boldsymbol{\nabla}) - h_\Gamma(-i\boldsymbol{\nabla})$ term is the analog to the particle-hole symmetry breaking term in the BM model for tBG [7], and is likewise suppressed by a factor $\mathcal{O}(\theta)$. If we drop this term in Eqn 3 by approximating $h_\Gamma(-i\mathcal{R}^{-1}_\theta\boldsymbol{\nabla}) \simeq h_\Gamma(-i\boldsymbol{\nabla})$, the resulting Hamiltonian gains a new symmetry $U = \exp(i\frac{\pi}{4}\sigma_2)$, where $\sigma_2$ is a Pauli matrix in the layer space. The action of $\psi_{\Gamma,l,\alpha} = U_{ll'}\psi'_{\Gamma,l',\alpha}$ admixes the two layers into a decoupled basis,

$$H_{\text{moire},\Gamma} = \sum_{\pm} \int d^2\mathbf{x}\, \psi'^{\dagger}_{\Gamma,\pm,\alpha}(\mathbf{x}) h^{\alpha\beta}_{\Gamma}(-i\boldsymbol{\nabla})\psi'_{\Gamma,\pm,\beta}(\mathbf{x}) \mp t\, \psi'^{\dagger}_{\Gamma,\pm,\alpha}(\mathbf{x}) \begin{pmatrix} 1 & 0 \\ 0 & 1 \end{pmatrix} \psi'_{\Gamma,\pm,\alpha}(\mathbf{x})$$

$$+ 2(w_{0,\text{os}} \mp w_{0,\text{t}})\, \psi'^{\dagger}_{\Gamma,\pm,\alpha}(\mathbf{x}) \begin{pmatrix} 1 & 0 \\ 0 & 1 \end{pmatrix} \Big[ \cos(\mathbf{q}_1 \cdot \mathbf{x}) + \cos(\mathbf{q}_2 \cdot \mathbf{x}) \Big] \psi'_{\Gamma,\pm,\alpha}(\mathbf{x})$$

$$+ 2(w_{1,\text{os}} \mp w_{1,\text{t}})\, \psi'^{\dagger}_{\Gamma,\pm,\alpha}(\mathbf{x}) \begin{pmatrix} 0 & 1 \\ 1 & 0 \end{pmatrix} \Big[ \cos(\mathbf{q}_1 \cdot \mathbf{x}) - \cos(\mathbf{q}_2 \cdot \mathbf{x}) \Big] \psi'_{\Gamma,\pm,\alpha}(\mathbf{x})$$

$$\equiv H_+ + H_- . \tag{4}$$

Either copy ($H_{\pm}$) of Eqn 4 is a layer bonding/anti-bonding sector which are each mathematically equivalent to a problem of a single monolayer experiencing an external superlattice, i.e of the aforementioned case (I) – see independent derivation in Sec 2.1. The copies are separated in energy by $2t$, where $t$ corresponds to the average moire tunneling over the entire system. Any mixing of these copies comes from the neglected $h_{\Gamma}(-i\mathcal{R}^{-1}_{\theta}\boldsymbol{\nabla}) - h_{\Gamma}(-i\boldsymbol{\nabla})$ contribution, which we found to be insignificant compared to $t$, the dominate contribution to the tunneling.

Likewise, the problem is further decoupled in spin, seeing that $\lambda$ only reduces the spin symmetry to $U(1)$. Therefor our purposes, it will be sufficient to focus on $H_-$ in the $\uparrow$-spin sector, with the understanding that $H_+$ is equivalent up to taking $w_{0/1,\text{t}} \rightarrow -w_{0/1,\text{t}}$; $\lambda \rightarrow -\lambda$ flips the spin; and the remaining constant $t$ can be removed by a constant shift in energy. We redefine $w_{0/1} \equiv (w_{0/1,\text{os}} + w_{0/1,\text{t}})$ and the 2-component spinor $\psi \equiv \psi'_{\Gamma,\uparrow}$, then write

$$H_{-,\uparrow} = \int d^2\mathbf{x}\, \psi^{\dagger}(\mathbf{x}) h^{\uparrow\uparrow}_{\Gamma}(-i\boldsymbol{\nabla})\psi(\mathbf{x}) + 2w_0\, \psi^{\dagger}(\mathbf{x}) \begin{pmatrix} 1 & 0 \\ 0 & 1 \end{pmatrix} \Big[ \cos(\mathbf{q}_1 \cdot \mathbf{x}) + \cos(\mathbf{q}_2 \cdot \mathbf{x}) \Big] \psi(\mathbf{x})$$

$$+ 2w_1\, \psi^{\dagger}(\mathbf{x}) \begin{pmatrix} 0 & 1 \\ 1 & 0 \end{pmatrix} \Big[ \cos(\mathbf{q}_1 \cdot \mathbf{x}) - \cos(\mathbf{q}_2 \cdot \mathbf{x}) \Big] \psi(\mathbf{x}) . \tag{5}$$

A key feature of this Hamiltonian is the aforementioned quadratic band touching, which occurs in the limit $\lambda = 0$ [70]. In this limit, turning on a small $w_1/(aq^2_{\mathcal{S}})$ opens a gap everywhere in the mBZ except for the quadratic band touching at $\Gamma$. Since the Hamiltonian is everywhere real valued in the absence of spin-orbit, turning on non-zero $\lambda$, and therefore gapping the node at $\Gamma$, guarantees a Chern number 1 (or $-1$ depending on the sign of $\lambda$) for the upper superlattice band. Because $\lambda$ only reduces the $SU(2)$ spin-symmetry down to $U(1)$, opposite spin sectors decouple within degenerate bands, which are necessarily degenerate and have opposite Chern number due to time-reversal [85].

As sketched in Fig 2a, keeping $\lambda \neq 0$ while tuning the ratio $w_0/w_1$ results in a transition into a $C = 0$ state, which occurs through a Dirac cone at the mBZ corner ($\mathbf{k} = \text{M}_m$) [86]. The topological phase boundary between $C = 0$ and 1 occurs when

$$\sqrt{(1 + 2\tilde{w}_1)^2 + \tilde{\lambda}^2} = \sqrt{1 + (\tilde{\lambda} + 2\tilde{w}_0)^2}, \tag{6}$$

for dimensionless $\tilde{\lambda} \equiv \lambda/(|a|q^2_{\mathcal{S}}/4)$, $\tilde{w}_{0/1} \equiv w_{0/1}/(|a|q^2_{\mathcal{S}}/4)$. Thus this transition can be tuned via the superlattice period $l_S = 2\pi/q_{\mathcal{S}}$.

Close to the phase boundary on the trivial side, the system is best described in terms of nearly free holes. However, spin-orbit coupling flattens the bands in the vicinity of $\Gamma$ (at which the Fermi velocity is zero), and since the gradients of the hole fields are cutoff by $q_{\mathcal{S}}$, smaller $q_{\mathcal{S}}$ guarantees flatter bands. In fact, if the free-hole bandwidth (set by $q_{\mathcal{S}}$) is much smaller than the $2\lambda$ splitting of the continuum bands, i.e if

$$2\tilde{\lambda} \gg \tilde{\lambda} + 2\left|\frac{\mu}{a}\right| - \sqrt{1 + \tilde{\lambda}^2}, \tag{7}$$

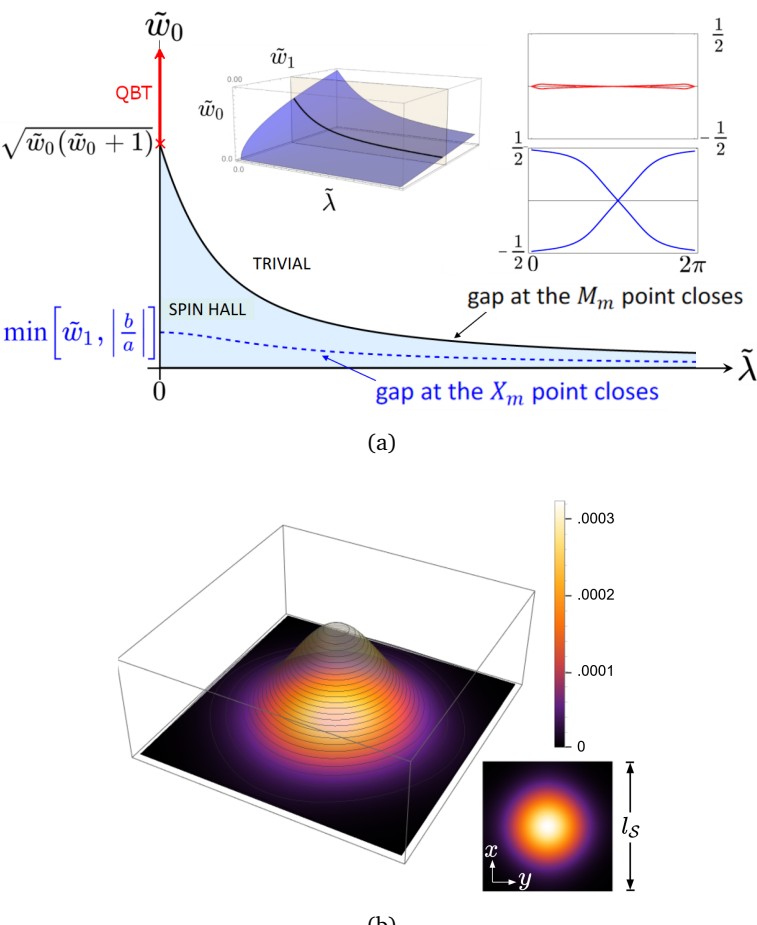

(a)

(b)

Figure 2: (a) Phase diagram for the upper moire hole band at fixed $\tilde{w}_1$. The boundary between the spin Hall and trivial phase corresponds to the black line in the 3D diagram (shown as inset). The charge center winding within the unit cell, as a function of mBZ momentum $k_y \in \{0, 2\pi\}$, is shown for the two phases. The quadratic-band touching at $\Gamma$ is preserved along the $\tilde{\lambda} = 0$ line. The red × indicates a triple-band touching at $M_m$, beyond which becomes a quadratic-band touching (shown as a red line). The diagram for $\tilde{w}_0 < 0$ is equivalent, except for the absence of the dashed blue line, because the gap at $X_m$ does not close there. (b) Density of the Wannier state for the band in the trivial phase, calculated via the projection method [87].

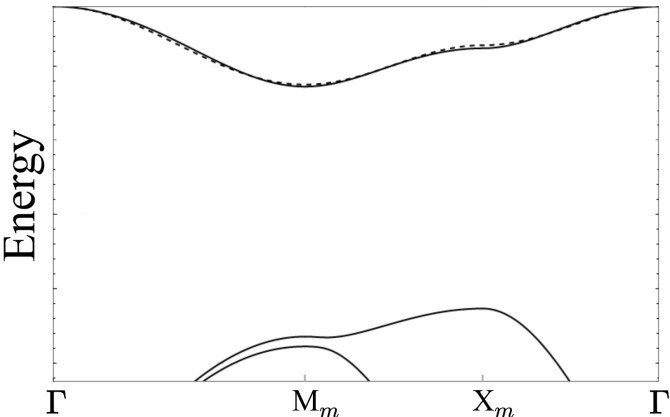

Figure 3: Lowest energy bands in the trivial phase. The top band corresponds to the Wannier functions shown in Fig 2b. The dashed line shows the fit to nearest-neighbor square lattice hopping dispersion $E_{\mathbf{k}} = -\epsilon\big(\cos(\frac{2\pi}{q_S}k_x) + \cos(\frac{2\pi}{q_S}k_y)\big)$.

we can safely project onto the upper continuum band. This amounts to seeking eigenstates of Eqn 5 of the form $\psi(\mathbf{x}) = \phi(\mathbf{x})(1, i)^T/\sqrt{2}$:

$$-\mu\nabla^2\phi + 2w_0\big(\cos(q_S x) + \cos(q_S y)\big)\phi = E\phi\,. \tag{8}$$

The potential of Eqn 8 has minima arranged in a crystal with period $l_S$,

$$-\mu\nabla^2\phi - 2w_0\frac{1}{2}q_S^2(x^2 + y^2)\phi \simeq (E - 2w_0)\phi\,. \tag{9}$$

Thus we can exactly calculate Wannier functions in this limit, being the eigenstates of a 2D harmonic oscillator with a frequency and mass defined by $m\omega^2/2 \equiv |w_0|q_S^2$ and $\hbar^2/(2m) \equiv |\mu|$. Their density is localized over a lengthscale

$$l = \frac{1}{\sqrt{q_S}}\left|\frac{\mu}{w_0}\right|^{\frac{1}{4}}, \tag{10}$$

which is either centered on the middle or corner of the unit cell, depending on the sign of $w_0$. The existence of localized Wannier states suggests a tight-binding description of the system, which is appropriate given a majority of the density lies within the unit cell, i.e $l \ll l_S$, or equivalently

$$\frac{1}{\sqrt{q_S}}\left|\frac{\mu}{w_0}\right|^{\frac{1}{4}} \ll \frac{2\pi}{q_S}\,. \tag{11}$$

This is guaranteed for sufficiently small $\theta$.

The two equations Eqn 7 & 11 set conditions on the validity of a tight-binding description of the system, beyond which the system is better described as nearly free holes:

$$\theta \ll \min\left[5.4^o, \left(31.9\sqrt{w_0[\text{meV}]}\right)^o\right], \tag{12}$$

for $w_0$ given in units of meV. The numbers for Eqn 12 have been fixed using our fits of the $\mathbf{k}\cdot\mathbf{p}$ invariants to photoemission [58]; as well as the experimentally determined 2-Fe unit cell lattice spacing $l_{\text{Fe}} = 3.77$ Å [33], which determines $l_S = l_{\text{Fe}}/\theta$ for a given $\theta$.

In Fig 2b, we plot the Wannier function for the trivial band (Fig 3), calculated numerically via the projection method [87], and find that they are well localized even away from the exact

$\theta \to 0$ limit. As $\theta \to 0$, the bands become increasingly flat, and their degeneracy structure approaches that of the 2D harmonic oscillator (Fig 4) with energy spacing

$$\hbar\omega = 2q_S \sqrt{|\mu|}\sqrt{|w_0|} \simeq 3.1 \times \theta \sqrt{w_0 [\text{meV}]}, \tag{13}$$

where $\theta$ here is taken in degrees. By comparison, the on-site Coulomb energy follows

$$V_{\text{Coulomb}} = \frac{\sqrt{\pi}}{2\sqrt{2}} \frac{e^2}{\epsilon l} \simeq \frac{211 \text{ meV}}{\epsilon} \times \sqrt{\theta} \, |w_0 [\text{meV}]|^{\frac{1}{4}}, \tag{14}$$

the prefactor for which becomes comparable to the 3.1 of Eqn 13 when we consider $\epsilon_{\text{SrTiO}_3} \geq 300$ [88,89].

Lastly, we fit

$$\left(w_{0,\text{os}}, w_{0,\text{t}}, |w_{1,\text{os}} + w_{1,\text{t}}|, |w_{1,\text{os}} - w_{1,\text{t}}|, t\right) = (3.4, -.9, .3, .6, 32.3) \text{ meV},$$

with the aid of DFT [72–80] by studying the spectrum of bilayer FeSe at $\mathbf{k} = \Gamma$ for three stacking configurations: $\mathbf{u}_{\text{AA}} = (0, 0)$, $\mathbf{u}_{\text{AB}} = (l_{\text{Fe}}/2, l_{\text{Fe}}/2)$, and $\mathbf{u}_{\frac{1}{2}} = (l_{\text{Fe}}/2, 0)$. This is done with an inter-layer distance $c = 5.345$ Å, which we found minimized the energy of AA stacked bilayers.

It should be pointed out however, that due to the renormalizations in the real material, DFT-determined values for the invariants may not be accurate. As such, we have tried to constrain ourselves to experiment as much as possible, and report the values of $w$'s determined via DFT with the understanding that they at best reflect an estimate of their order of magnitude. Nevertheless, the sizeable $\lambda$ reported by experiments, in combination with large $w_0/w_1$ expected from DFT, favours Hubbard-type physics for the moire bands about $\Gamma$.

It has been shown that pressure facilitates the moire physics in tBG [90, 91], which is generally expected since pushing the layers closer together increases the inter-layer tunneling strength. Since the trivial bands described here flatten with decreasing $\theta \propto l_S^{-1}$, interaction-driven phases are not tied to a specific magic angle. Therefore the combination of both $\theta$ and pressure provides a tunable multidimensional space of achievable device configurations, which could be used to explore strong-coupling physics.

Likewise, it remains to be seen what role moire ferroelectrics will have in engineering superlattice substrates [92, 93]. For example, twisted hexaboron nitrides have been shown to produce a triangular superlattice potential larger than 200 meV [92], which if placed atop an electron/hole-carrying layer, would modulate the on-site energy of those carriers, and therefore act as an external superlattice. As we have pointed out, small twist-angle moire physics at $\Gamma$ (type II) is equivalent to a system of a single monolayer experiencing a square superlattice (type I). Such devices may be easier to construct and tune, and are not constrained by the natural size and ratios of the tunnelings between relaxed homobilayers. One possible route to a square superlattice would be to twist (or misalign) FeSe against ultra-thin films of $\text{BiFeO}_3$, which are by themselves ferroelectric [94].

The purpose of this manuscript is two fold: lay down the general theoretical framework necessary for future investigations into a new class of moire heterostructures, as well as provide a clear proposal for immediate experimental exploration of a specific device composed of either $\text{FeSe}/\text{SrTiO}_3$ [26, 95–97] or $\text{FeTe}/\text{SrTiO}_3$ [52, 55] monolayers. To this end we have organized this paper in such a way as to prioritize the experimental proposal laid out in this Intro, starting first with details specific to moire bands at $\Gamma$. This includes a from-microscopics derivation of the moire tunneling in Eqn 3 in Sec 2.1, which is followed by a detailed analysis of the superlattice band structure in Sec 2.2. (In the appendix we analyze a second exact mathematical limit of the model, which may be achievable in materials with a quadratic-band touching, but where the particle-hole asymmetry is not too large.) We then pivot in

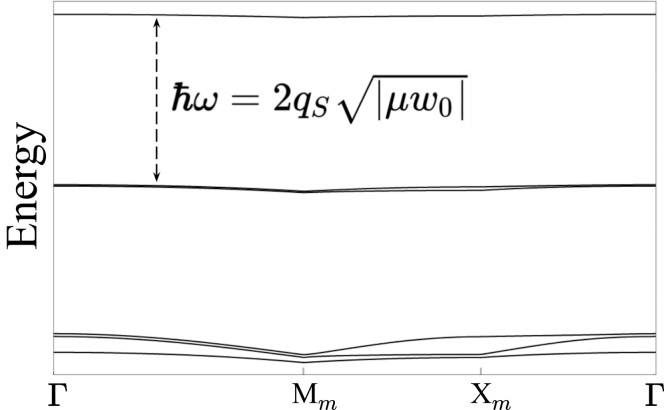

Figure 4: Energy of the bands in the tight-binding limit, which occurs deep in the trivial phase of Fig 2a. The degeneracy structure approaches: $1, 2, 3, ..., n, ...$ for the $n^{\text{th}}$ set of bands.

the following sections into an in-depth discussion of space group symmetries, starting with the irreducible representations at $\Gamma$ and $M$ in Sec 3.1. In Sec 3.2, we derive a general form for both the moire superlattice tunnelings and superlattice potentials, and discuss how they transform under space group symmetries. In sections 3.3-3.6 we use symmetry to derive said tunnelings/potentials at both high symmetry points. Our discussion of bilayer FeSe ends with a detailed explanation of how we fit our tunnelings/potentials using DFT in Sec 3.7. Finally, we conclude with a discussion of the theoretical and experimental possibilities that this work opens by tying together the fields of moire physics and Fe-based superconductivity.

## 2 Moire bands at $\Gamma$

### 2.1 Microscopic derivation of the inter-layer tunneling for the fields near $\Gamma$

Let us consider two stacked infinitely-large monolayers, which are relatively (and uniformly – see Ref [83] for a more general derivation in graphene) rotated by an angle $\theta \ll 1$. For what follows, let $a$ index the combination of sublattice and orbital (e.g $a = (Xz, A)$ for an $d_{Xz}$-orbital at sublattice $A$). WLOG, we consider an unrotated lab coordinate $\mathbf{x} \in \{n(l_{\text{Fe}}, 0) + m(0, l_{\text{Fe}}) | (n, m) \in \mathbb{Z}^2\}$ in which one layer is fixed $\mathbf{r}_1 = \mathbf{x}$, and the other rotated $\mathbf{r}_2 = R_\theta \mathbf{x}$; and similarly for the loci of their sublattices $\boldsymbol{\delta}_a$. The general form for our uniformly rotated bilayers in the tight-binding limit looks like

$$H_{\text{t.b}} = \sum_{\mathbf{x}, \mathbf{x}'} \sum_{a,b} d_{a,1}^\dagger(\mathbf{x}) t_{ab}(\mathbf{x} - R_\theta \mathbf{x}' + \boldsymbol{\delta}_a - R_\theta \boldsymbol{\delta}_b + c\hat{\mathbf{z}}) d_{b,2}(R_\theta \mathbf{x}') + \text{h.c.} \tag{15}$$

Note that the overlaps $t_{ab}$ are computed for bilayers which are separated in the $z$-direction by an inter-layer distance $c$. Since we do not consider corrugation effects, $c$ is constant. We also consider the indices $a$ & $b$ to span the same basis of atomic orbitals defined with respect to the unrotated frame. As a consequence, $t_{ab}$ has an implicit $\theta$ dependence necessary to account for the shape of the tilted Wannier functions. However, at small angles, the rotated Wannier functions can be understood to be perturbatively connected to their unrotated counterparts by $\mathcal{O}(\theta)$, and therefore, the correction to the tunneling due to this tilt in their anisotropy is likewise $\mathcal{O}(\theta)$. Since we work within the assumption where $\mathcal{O}(\theta)$ tunneling is sub-leading, as it will be for small enough $\theta$, we suppress it here.

We will henceforth proceed toward the effective theory for the fields about the high symmetry points. Because the system is infinitely large, there are uncountably many single-particle states in the atomic Brillouin zone, which is reflected as an integral $\int_{\mathbf{k}} \equiv \int \frac{d^2\mathbf{k}}{(2\pi)^2}$. Moving into momentum space has the caveat that the rotated layer's Brillouin zone is likewise rotated, due to the tilted axis of its periodicity. Rather than integrate over a rotated Brillouin zone, we choose to Fourier transform the second layer like

$$d_{a,2}(R_\theta \mathbf{x}') = \int_{\mathbf{k}} e^{iR_\theta \mathbf{x}' \cdot R_\theta \mathbf{k}} d_{a,2}(R_\theta \mathbf{k}), \tag{16}$$

and span the $\mathbf{k}$ integral over the unrotated zone. (Note that $R_\theta \mathbf{x} \cdot R_\theta \mathbf{k} = \mathbf{x} \cdot \mathbf{k}$.) For consistency, for the rest of this calculation, we continue with this practice of summing/integrating over the unrotated frame.

Moving into the effective theory amounts to a projection not only onto the relevant bands, but also onto the states which are in the vicinity of the high symmetry points. I.e keeping only those $d_a(\mathbf{k} + \mathcal{Q})$ from the expansion of $d_a(\mathbf{x})$ where $|\mathbf{k}| \ll 2\pi/l_{\text{Fe}}$ and $\mathcal{Q} \in \{\Gamma, M\}$:

$$\simeq \sum_{\mathcal{Q}, \mathcal{Q}'} \int_{\mathbf{k}, \mathbf{k}'} \int_{\mathbf{p} \in \mathbb{R}^2} d_{a,1}^\dagger(\mathbf{k} + \mathcal{Q}) t_{ab}(\mathbf{p}) d_{b,2}(R_\theta \mathbf{k}' + R_\theta \mathcal{Q}') \int_{\mathbf{x}} e^{i\mathbf{x} \cdot (\mathbf{p} - \mathbf{k} - \mathcal{Q})} \int_{\mathbf{x}'} e^{-i\mathbf{x}' \cdot (R_\theta^{-1} \mathbf{p} - \mathbf{k}' - \mathcal{Q}')}$$
$$+ \text{h.c.} \tag{17}$$

Where "$\simeq$" here reflects the fact that the terms lost in the projection are at higher energy (and therefore less relevant to the low energy theory we are interested in constructing). The transition from $\sum_{\mathbf{x}} \to \int d^2\mathbf{x} \equiv \int_{\mathbf{x}}$ reflects the move into the continuum limit, which is valid for $|\mathbf{k}| \ll 2\pi/l_{\text{Fe}}$. However, we should remember that the integral over $\mathbf{x}$ originates from a sum over atomic lattice points, and therefore $e^{i\mathbf{r}_l \cdot \mathbf{Q}_{(l)}} = 1$ for $\mathbf{Q}_{(l)} \equiv \delta_{l,1}\mathbf{Q} + \delta_{l,2}R_\theta \mathbf{Q}$; where $\mathbf{Q} \in \{n(2\pi/l_{\text{Fe}}, 0) + m(0, 2\pi/l_{\text{Fe}}) | (n, m) \in \mathbb{Z}^2\}$. Thus

$$\int_{\mathbf{x}} e^{i\mathbf{x} \cdot (\mathbf{p} - \mathbf{k} - \mathcal{Q})} \int_{\mathbf{x}'} e^{-i\mathbf{x}' \cdot (R_\theta^{-1} \mathbf{p} - \mathbf{k}' - \mathcal{Q}')} = \sum_{\mathbf{Q}, \mathbf{Q}'} \delta(\mathbf{p} - \mathbf{k} - \mathcal{Q} - \mathbf{Q}) \delta(\mathbf{p} - R_\theta \mathbf{k}' - R_\theta \mathcal{Q}' - R_\theta \mathbf{Q}') \tag{18}$$

demands $\mathbf{k} + \mathcal{Q} + \mathbf{Q} = R_\theta \mathbf{k}' + R_\theta \mathcal{Q}' + R_\theta \mathbf{Q}'$ for some (unrotated) reciprocal lattice vectors $\mathbf{Q}, \mathbf{Q}'$. Since $|\mathbf{k} - R_\theta \mathbf{k}'| \ll 2\pi/l_{\text{Fe}}$ and $\theta \ll 1$, then individually both $\mathcal{Q}' = \mathcal{Q}$ and $\mathbf{Q}' = \mathbf{Q}$. The prior has the meaning that the scattering between different high symmetry points is suppressed. What remains is the scattering within a high symmetry point,

$$= \sum_{\mathcal{Q}} \int_{\mathbf{k}} d_{a,1}^\dagger(\mathbf{k} + \mathcal{Q}) \sum_{\mathbf{Q}} t_{ab}(\mathbf{k} + \mathcal{Q} + \mathbf{Q}) d_{b,2}(\mathbf{k} + \mathcal{Q} + \mathbf{Q} - R_\theta \mathbf{Q}) + \text{h.c.} \tag{19}$$

The tunneling $t_{ab}(\mathbf{k} + \mathcal{Q} + \mathbf{Q})$ is analytic in $\mathbf{k}$. The $\mathbf{k} = 0$ part $t_{ab}(\mathcal{Q} + \mathbf{Q})$ is the leading order term of the expansion in small $|\mathbf{k}| \ll 2\pi/l_{\text{Fe}}$. We keep only the $\mathbf{k}$-independent tunneling:

$$H_{\text{t.b}} \simeq \sum_{\mathcal{Q}} \int_{\mathbf{k}} d_{a,1}^\dagger(\mathbf{k} + \mathcal{Q}) \sum_{\mathbf{Q}} t_{ab}(\mathcal{Q} + \mathbf{Q}) d_{b,2}(\mathbf{k} + \mathcal{Q} + \mathbf{Q} - R_\theta \mathbf{Q}) + \text{h.c} \equiv H_{\text{eff}}. \tag{20}$$

Recognize that $\mathbf{q} = \mathbf{Q} - R_\theta \mathbf{Q}$ are the moire reciprocal lattice vectors. The map between $\mathbf{Q}$ and $\mathbf{q}$ is reversible, such that there is one $\mathbf{Q}$ for every $\mathbf{q}$, and thus we can relabel our $t_{ab}$ in terms of $\mathbf{q}$: $t_{ab}(\mathbf{Q}) \to t_{ab}(\mathbf{q})$.

Generically, in order to construct an effective theory for FeSe, we would need to consider $Xz$, $Yz$, $xy$ orbitals at both sublattices in the 2-Fe unit cell. However, this construction is

greatly simplified if we are only interested in $\mathcal{Q} = \Gamma$ – then only $Yz$ and $Xz$ orbitals at a single sub-lattice are necessary.

$$H_\Gamma = \int_{\mathbf{k}} d^\dagger_{a,1}(\mathbf{k}) \sum_{\mathbf{q}} t_{ab}(\mathbf{q}) d_{b,2}(\mathbf{k}+\mathbf{q}) + \text{h.c.} \tag{21}$$

We further constrain ourselves to $|\mathbf{q}| \leq |\mathbf{q}_1|$, which correspond to the longest wavelength scattering. This can be understood intuitively as being the constraint that the inter-layer tunneling is dominated by the slowest variations in the moire pattern. But the specific reason for this is due to $t_{ab}(r)$ being a function of the 3D displacement $r = \sqrt{r_\parallel^2 + c^2}$, which varies in $r_\parallel$ as

$$\frac{d t_{ab}(r)}{d r_\parallel} = \frac{dr}{d r_\parallel} \frac{d t_{ab}(r)}{dr} = \frac{r_\parallel}{\sqrt{r_\parallel^2 + c^2}} \frac{d t_{ab}(r)}{dr}, \tag{22}$$

and hence changes slowly on the scale of $r_\parallel < c$. In other words, the hopping function is less sensitive to variations on length scales below $c$. This in turn means that the in-plane gradient of the tunneling, and therefore the scattering momenta $q$, are suppressed for $q > 1/c$ [4].

Despite the expectation that the tunneling falls off for large $\mathbf{q}$, nothing a priori tells us to keep *only* $|\mathbf{q}| \leq |\mathbf{q}_1|$. Higher order contributions to the tunneling may play a sub-dominate roll, however, we have no way of knowing how much at this time. Since the superlattice band gap (discussed in main text) is set by the leading order tunneling, any introduced sub-leading terms would have to overcome that energy scale, and thus close the gap, in order to fundamentally restructure the bands. Therefore, we expect that the $|\mathbf{q}| \leq |\mathbf{q}_1|$ tunneling is sufficient to capture the topological nature of the bands.

Along this line, one should expect the zeroth order term $t_{ab}(\mathbf{0}) \equiv t \delta_{ab}$ to be the dominate contribution from the tunneling – we confirmed this using DFT. It represents the average tunneling over the entire moire crystal, and by itself preserves the translational symmetry of the continuum theory. The leading order "pattern part" of the potential are those with $|\mathbf{q}| = |\mathbf{q}_1|$:

$$\begin{aligned}H_\Gamma = \int_{\mathbf{k}} d^\dagger_{a,1}(\mathbf{k})\Big(t d_{a,2}(\mathbf{k}) + t_{ab}(\mathbf{q}_1)d_{b,2}(\mathbf{k}+\mathbf{q}_1) + t_{ab}(-\mathbf{q}_1)d_{b,2}(\mathbf{k}-\mathbf{q}_1) \\ + t_{ab}(\mathbf{q}_2)d_{b,2}(\mathbf{k}+\mathbf{q}_2) + t_{ab}(-\mathbf{q}_2)d_{b,2}(\mathbf{k}-\mathbf{q}_2)\Big) + \text{h.c.}\end{aligned} \tag{23}$$

Because $a \in \{Yz, Xz\}$, each $t_{ab}$ is a $2 \times 2$ matrix. We wish to know which of these 16 hopping processes between the stated $d$ orbitals are not allowed by the symmetry of the $Yz/Xz$ orbitals, and if any could be related. We illustrate the relationship between the orientation of the orbital and the direction of the scattering in Fig 5, from which one can deduce that all $t_{a,b}(\mathbf{q}_j)$ where $a = b$ are equal – we called this $w_{0,t}$ in the main text. And $t_{Yz,Xz}(\mathbf{q}_1) = t_{Yz,Xz}(-\mathbf{q}_1)$, but $t_{Yz,Xz}(\mathbf{q}_1) = -t_{Yz,Xz}(\mathbf{q}_2)$ – which we call $w_{1,t}$. Additionally, time reversal guarantees both $t_{Yz,Yz}(\mathbf{q}_1) \equiv w_{0,t}$ and $t_{Yz,Xz}(\mathbf{q}_1) \equiv w_{1,t}$ are real valued:

$$\begin{aligned}H_\Gamma = \int_{\mathbf{k}} d^\dagger_{a,1}(\mathbf{k})\Big(t d_{a,2}(\mathbf{k}) + w_0\big[d_{a,2}(\mathbf{k}+\mathbf{q}_1) + d_{a,2}(\mathbf{k}-\mathbf{q}_1) + d_{a,2}(\mathbf{k}+\mathbf{q}_2) + d_{a,2}(\mathbf{k}-\mathbf{q}_2)\big]\Big) \\ + w_1\Big(d^\dagger_{Yz,1}(\mathbf{k})\big[d_{Xz,2}(\mathbf{k}+\mathbf{q}_1) + d_{Xz,2}(\mathbf{k}-\mathbf{q}_1) - d_{Xz,2}(\mathbf{k}+\mathbf{q}_2) - d_{Xz,2}(\mathbf{k}-\mathbf{q}_2)\big] \\ + d^\dagger_{Xz,1}(\mathbf{k})\big[d_{Yz,2}(\mathbf{k}+\mathbf{q}_1) + d_{Yz,2}(\mathbf{k}-\mathbf{q}_1) - d_{Yz,2}(\mathbf{k}+\mathbf{q}_2) - d_{Yz,2}(\mathbf{k}-\mathbf{q}_2)\big]\Big) \\ + \text{h.c.}\end{aligned} \tag{24}$$

We can then produce Eqn 3 by defining the Fourier transform

$$\hat{d}_{a,l}(\mathbf{x}) = \int_{\mathbf{k}} d_{a,l}(\mathbf{k}) e^{i\mathbf{x}\cdot\mathbf{k}}. \tag{25}$$

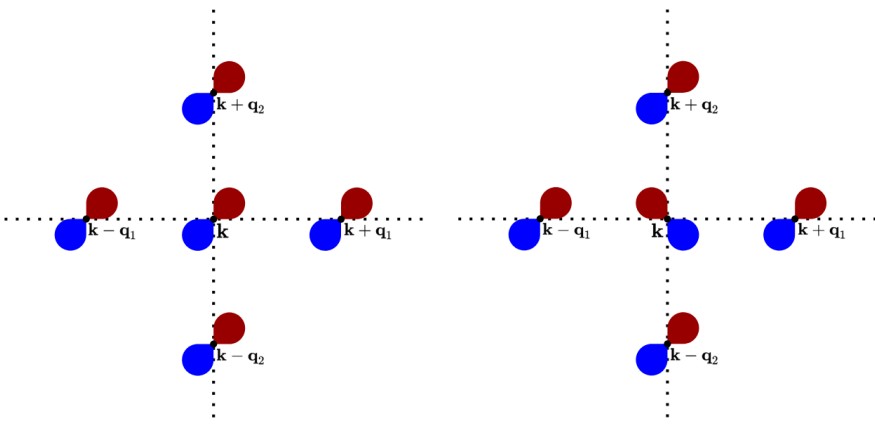

Figure 5: Tunelling between orbitals in neighboring superlattice BZ's: $T(\mathbf{q}_j)_{Yz,Yz}$ (left) and $T(\mathbf{q}_j)_{Yz,Xz}$ (right). The red/blue teardrops represent the d-wave symmetry of the atomic orbitals, as seen from above.

The additional "hat" implies that $\hat{d}_{a,l}(\mathbf{x})$ has all modes away from the $\Gamma$ point projected out, and are therefore not the original localized orbitals $d_{a,l}(\mathbf{x})$. They do however share their anisotropic character, transforming like $Yz/Xz$ orbitals, such that the only overlaps which survived are those which combined with the plane waves to produce invariants.

## 2.2 Folding bands and opening gaps in the superlattice Brillouin zone

The effective kinetic energy Eqn 2 has an emergent continuous translational symmetry, such that its Fourier transform

$$
\begin{aligned}
h_\Gamma(\mathbf{p})\psi_\Gamma(\mathbf{p}) &= \left(\epsilon_\Gamma + \mu p^2 + a p_x p_y \tau_3 + b(p_x^2 - p_y^2)\tau_1 + \lambda \tau_2\right)\psi_\Gamma(\mathbf{p}) \\
&\equiv \left(h_0(\mathbf{p}) + h_3(\mathbf{p})\tau_3 + h_1(\mathbf{p})\tau_1 + h_2\tau_2\right)\psi_\Gamma(\mathbf{p})
\end{aligned}
\tag{26}
$$

is defined over the domain $\mathbf{p} \in \mathbb{R}^2$ $\left(\text{For later convenience, we define } h(\mathbf{p}) \equiv \sqrt{h_1^2 + h_2^2 + h_3^2}\right)$. WLOG due to time-reversal symmetry, let $\lambda \geq 0$. In moving to the superlattice picture, we relabel $\mathbf{p} = \mathbf{k} + \mathbf{q}$, where $\mathbf{q} = \{n\mathbf{q}_1 + m\mathbf{q}_2|(n,m) \in \mathbb{Z}^2\}$ and $\mathbf{k} \in$ mBZ. The inter-layer term in Eqn 27 is translationally invariant under superlattice shifts, and thus the full Hamiltonian is labelled by crystal momentum $\mathbf{k}$:

$$
H_\mathbf{k}\psi(\mathbf{k}-\mathbf{q}) = h_\Gamma(\mathbf{k}-\mathbf{q})\psi(\mathbf{k}-\mathbf{q})
\tag{27}
$$

$$
+ \sum_{\mathbf{q}'}\sum_{\pm}\left[\begin{pmatrix} w_0 & w_1 \\ w_1 & w_0 \end{pmatrix}\delta_{\mathbf{q}'\pm\mathbf{q}_1,\mathbf{q}} + \begin{pmatrix} w_0 & -w_1 \\ -w_1 & w_0 \end{pmatrix}\delta_{\mathbf{q}'\pm\mathbf{q}_2,\mathbf{q}}\right]\psi(\mathbf{k}-\mathbf{q}').
\tag{28}
$$

The superlattice is square, so any mBZ centered at $\mathbf{q}$ shares a boundary with the adjacent mBZ's at $\mathbf{q}\pm\mathbf{q}_1$ and $\mathbf{q}\pm\mathbf{q}_2$. In the absence of $w_{0,1}$, band crossings exists at these boundaries due to the folding of the continuum bands into the superlattice zone: along $\overline{M_m X_m}$ and $\overline{M_m Y_m}$, as well as a four-band crossing at the mBZ corner $M_m$. It is these band crossings that the action of $w_{0,1}$ needs to open in order for the upper band to be separated from the remote bands everywhere except at $\Gamma$. The quadratic band touching at $\Gamma$ is gapped by $\lambda$.

### 2.2.1 Gap between adjacent mBZ's: $\overline{XM_m}$ and $\overline{YM_m}$

The plane wave basis is infinite, giving rise to an infinite number of bands at every $\mathbf{k}$. As a practical matter, one can approximate Eqn 27 about a given $\mathbf{k}_0$ by truncating the number of plane

waves to a chosen subset of $\mathbf{q}$ which minimize $|\mathbf{q} - \mathbf{k}_0|$. In the limit of small tunnelings, this truncation can be made accurate for a given band, so long as the energy difference between that bands and the truncated bands is finite when $w_0 = w_1 = 0$. In other words, we can approximate the top band about point $\mathbf{k}_0 \in \overline{M_m X_m}$, accurately for small $\sqrt{(w_0)^2 + (w_1)^2}/|aq_{S^2}| \ll 1$, by truncating to that subset of bands with which at $\mathbf{k}_0$ it is degenerate in the absence of tunneling.

We start by expanding Eqn 27 in the basis of the two bands which cross along $\overline{X_m M_m}$, i.e $(k_x, k_y) = (q_S/2, k_y)$ and $k_y \neq q_S/2$. Since Eqn 26 is non-diagonal at X, we need to start with the four component spinor

$$\Psi_X(\mathbf{k}) = \left( \psi_\Gamma(\mathbf{k}), \psi_\Gamma(-\mathbf{q}_1 + \mathbf{k}) \right)^T, \tag{29}$$

in which basis Eqn 27 along $\overline{XM_m}$ becomes

$$H\left(\frac{q_S}{2}, k_y\right)\Psi_X\left(\frac{q_S}{2}, k_y\right) = \begin{pmatrix} h_\Gamma\left(\frac{q_S}{2}, k_y\right) & \hat{\delta} \\ \hat{\delta} & h_\Gamma\left(-\frac{q_S}{2}, k_y\right) \end{pmatrix}\Psi_X\left(\frac{q_S}{2}, k_y\right), \tag{30}$$

where

$$\hat{\delta} \equiv \begin{pmatrix} w_0 & w_1 \\ w_1 & w_0 \end{pmatrix}. \tag{31}$$

Rotating Eqn 30 into the band basis would produce two doubly degenerate pairs of bands, which are the eigenvalues of $h_\Gamma$ with energies

$$\epsilon_\pm(k_y) = \epsilon_\Gamma + \mu(q_S^2/4 + k_y^2) \pm \sqrt{(aq_S k_y/2)^2 + b^2(q_S^2/4 - k_y^2)^2 + \lambda^2}.$$

We do not yet know if the entire $\overline{XM_m}$ line is gapped for small $w$'s. We can determine this by projecting onto the upper degenerate pair. Note the following symmetry of Eqn 26: $h_\Gamma(\frac{q_S}{2}, k_y) = \tau_1 h_\Gamma(-\frac{q_S}{2}, k_y)^* \tau_1$. It then follows that the upper two bands have the following wavefunctions:

$$|\epsilon_+, q_S\rangle = \frac{(h_1 - ih_2, h - h_3)^T}{\sqrt{2h(h - h_3)}}, \tag{32}$$

$$|\epsilon_+, -q_S\rangle = \tau_1 |\epsilon_+, q_S\rangle^*. \tag{33}$$

We use these to define a projector onto the band basis $\mathcal{P}_{\epsilon_+} = \text{diagonal}(|\epsilon_+, q_S\rangle, |\epsilon_+, -q_S\rangle)$. Eqn 30 becomes

$$\mathcal{P}_{\epsilon_+}^\dagger H(\frac{q_S}{2}, k_y)\mathcal{P}_{\epsilon_+} = \begin{pmatrix} \epsilon_+(k_y) & \delta(k_y) \\ \delta^*(k_y) & \epsilon_+(k_y) \end{pmatrix}. \tag{34}$$

Even though $\hat{\delta}$ was real-valued and constant, it picks up a $k_y$-dependence upon projection,

$$\delta(k_y) = \langle \epsilon_+, q_S | \hat{\delta} | \epsilon_+, -q_S \rangle \tag{35}$$

$$= w_0 \frac{h_1}{h} - w_1 \left(1 - \frac{h_2^2}{h(h - h_3)}\right) + i\left(w_0 \frac{h_2}{h} - w_1 \frac{h_1 h_2}{h(h - h_3)}\right). \tag{36}$$

In order for the gap to close, both the real and imaginary parts of $\delta$ must equal zero simultaneously. This leads us to the set of equations

$$\frac{w_0}{w_1} = -\frac{h_3}{h_1} + \frac{h_1}{h - h_3} = \frac{h_1}{h - h_3}, \tag{37}$$

which only holds true when $k_y = 0$, in other words, at the high symmetry point $\mathbf{k} = X_m$. This is expected, because $\delta(k_y)$ inherits its momentum dependence from the bands of $h_\Gamma$, the components of which only have nodes along the high symmetry lines which cross $\Gamma$.

We can now return to $\mathbf{k} = X_m$ and re-solve for the eigenvalues of the four bands of Eqn 27. The Hamiltonian for which has the form

$$H_X = \left(\epsilon_\Gamma + \mu \frac{q_S^2}{4}\right) + b\frac{q_S^2}{4}\tau_1 + \lambda\tau_2 + \left(w_0 + w_1\tau_1\right)\Sigma_1,\tag{38}$$

where both $\tau_j$ and $\Sigma_j$ are Pauli matrices. Because $H_X$ depends on no other $\Sigma_j$ save $\Sigma_1$, choosing a basis for which $\Sigma_1$ is diagonal decouples the Hamiltonian into simpler subsectors: $H_{X,\pm} = \left(\epsilon_\Gamma + \mu\frac{q_S^2}{4}\right) + b\frac{q_S^2}{4}\tau_1 + \lambda\tau_2 \pm \left(w_0 + w_1\tau_1\right)$. It then follows that the four eigenvalues at X are

$$E_{X,++} = \epsilon_\Gamma + \mu\frac{q_{S^2}}{4} + w_0 + \sqrt{\left(b\frac{q_{S^2}}{4} + w_1\right)^2 + \lambda^2},\tag{39}$$

$$E_{X,+-} = \epsilon_\Gamma + \mu\frac{q_{S^2}}{4} - w_0 + \sqrt{\left(b\frac{q_{S^2}}{4} - w_1\right)^2 + \lambda^2},\tag{40}$$

$$E_{X,-+} = \epsilon_\Gamma + \mu\frac{q_{S^2}}{4} + w_0 - \sqrt{\left(b\frac{q_{S^2}}{4} + w_1\right)^2 + \lambda^2},\tag{41}$$

$$E_{X,--} = \epsilon_\Gamma + \mu\frac{q_{S^2}}{4} - w_0 - \sqrt{\left(b\frac{q_{S^2}}{4} - w_1\right)^2 + \lambda^2}.\tag{42}$$

Note that the set of four eigenvalues is invariant under $(w_0, w_1) \to (-w_0, -w_1)$, but not necessarily under the change of the sign of either $w$ individually. Therefore considering $w_1 > 0$, the gap to the upper band occurs when

$$w_0 + \sqrt{\left(b\frac{q_{S^2}}{4} + w_1\right)^2 + \lambda^2} = -w_0 + \sqrt{\left(b\frac{q_{S^2}}{4} - w_1\right)^2 + \lambda^2}.\tag{43}$$

Or equivalently,

$$2w_0 = \sqrt{\left(b\frac{q_{S^2}}{4} + w_1\right)^2 + \lambda^2} - \sqrt{\left(b\frac{q_{S^2}}{4} - w_1\right)^2 + \lambda^2},\tag{44}$$

which is valid only when $w_0 > 0$ & $w_1 > 0$, or $w_0 < 0$ & $w_1 < 0$. If we take $w_1 \to -w_1$, the right hand side of the equation changes sign, which is to say that the gap only closes in diagonal quadrants of the $(w_0, w_1)$ plane.

The gap along $\overline{Y_m M_m}$ is identical to that along $\overline{X_m M_m}$, being related by the mirror symmetry $m_X$. It should be noted that the gap at $M_m$ (discussed in the following section) and the gap at $X_m$ only precisely close together when spin-orbit is the dominate scale; otherwise, there exists a small window between their closing. Since the gap at $X_m$ closes and opens in the absence of spin-orbit, i.e when the Hamiltonian is everywhere real-valued, it cannot be of topological origin. We additionally verified this via a numerical Wilson loop calculation: that the closing of the gap at $X_m$ and $Y_m$ produces no change in the Chern number of the lowest energy band. Thus only the quadratic band touching at $\Gamma$, and the four-fold band crossing at $M_m$ play a role in the topological transition.

### 2.2.2 Gap at mBZ corner: $M_m$

Eqn 30 is not valid at $k_y = q_S/2$ (i.e at $M_m$) due to the larger degeneracy there. Thus we need to expand our truncated basis in order to account for the degeneracy.

$$\Psi_{M_m} = \big(\psi_{\Gamma,+}(M_m), \psi_{\Gamma,-}(M_m), \psi_{\Gamma,+}(-\mathbf{q_1}-\mathbf{q_2}+M_m), \psi_{\Gamma,-}(-\mathbf{q_1}-\mathbf{q_2}+M_m),$$

$$\psi_{\Gamma,-}(-\mathbf{q_1}+M_m), \psi_{\Gamma,+}(-\mathbf{q_1}+M_m), \psi_{\Gamma,-}(-\mathbf{q_2}+M_m), \psi_{\Gamma,+}(-\mathbf{q_2}+M_m)\big)^T. \quad (45)$$

The Hamiltonian in this expanded basis becomes

$$H_{M_m}\Psi_{M_m} = \begin{pmatrix} \epsilon & -i\lambda & 0 & 0 & w_1 & w_0 & -w_1 & w_0 \\ i\lambda & -\epsilon & 0 & 0 & w_0 & w_1 & w_0 & -w_1 \\ 0 & 0 & \epsilon & -i\lambda & -w_1 & w_0 & w_1 & w_0 \\ 0 & 0 & i\lambda & -\epsilon & w_0 & -w_1 & w_0 & w_1 \\ w_1 & w_0 & -w_1 & w_0 & \epsilon & i\lambda & 0 & 0 \\ w_0 & w_1 & w_0 & -w_1 & -i\lambda & -\epsilon & 0 & 0 \\ -w_1 & w_0 & w_1 & w_0 & 0 & 0 & \epsilon & i\lambda \\ w_0 & -w_1 & w_0 & w_1 & 0 & 0 & -i\lambda & -\epsilon \end{pmatrix} \Psi_{M_m} + (\epsilon_0 - \epsilon)\mathbb{1}_{4\times 4}\Psi_{M_m}, \quad (46)$$

where $\epsilon_0 \equiv \epsilon_\Gamma + \mu q_S^2/2 + a q_S^2/4$ and $\epsilon \equiv a q_S^2/4$. Dropping the term proportional to $\mathbb{1}_{4\times 4}$, the Hamiltonian takes the form

$$H_{M_m} = \epsilon\tau_3 + \lambda\tau_2\Sigma_3 + w_1(1-\sigma_1)\Sigma_1 + w_0\tau_1(1+\sigma_1)\Sigma_1. \quad (47)$$

Of the Pauli matrices $\sigma_j$, the Hamiltonian $H_{M_m}$ depends only on $\sigma_1$, which means we can rotate into a basis in which it is diagonal. This decouples the Hamiltonian into two sectors:

$$H_{M_m,\pm} = \epsilon\tau_3 + \lambda\tau_2\Sigma_3 + w_1(1\mp 1)\Sigma_1 + w_0\tau_1(1\pm 1)\Sigma_1, \quad (48)$$

or written less succinctly,

$$H_{M_m,+} = \epsilon\tau_3 + \lambda\tau_2\Sigma_3 + 2w_0\tau_1\Sigma_1, \quad (49)$$

$$H_{M_m,-} = \epsilon\tau_3 + \lambda\tau_2\Sigma_3 + 2w_1\Sigma_1. \quad (50)$$

We can determine the four eigenvalues of $H_{M_m,+}$ by first squaring it to show

$$H_{M_m,+}^2 - (\epsilon^2 + \lambda^2 + 4w_0^2) = -4w_0\lambda\tau_3\Sigma_2,$$

the right hand side of which is readily diagonalizable $-4w_0\lambda\tau_3\Sigma_2 \rightarrow \pm 4|w_0|\lambda$. Writing $\xi, \eta \in \{+1, -1\}$, the eigenvalues are

$$E_{+,\xi,\eta} = \xi\sqrt{\epsilon^2 + (\lambda + \eta 2|w_0|)^2}. \quad (51)$$

The same procedure can be applied to solve for the remaining four eigenvalues of $H_{M_m,-}$, which are

$$E_{-,\xi,\eta} = \xi\sqrt{(|\epsilon| + \eta 2|w_1|)^2 + \lambda^2}. \quad (52)$$

WLOG, let us consider $\epsilon, w_0, w_1 \geq 0$ (and remember $\lambda \geq 0$). The highest eigenvalues are $\sqrt{\epsilon^2 + (\lambda + 2w_0)^2}$ & $\sqrt{(\epsilon + 2w_1)^2 + \lambda^2}$. The spin Hall phase occurs when $\sqrt{(\epsilon + 2w_1)^2 + \lambda^2} > \sqrt{\epsilon^2 + (\lambda + 2w_0)^2}$, else we are in the trivial phase. There exists the quadratic-band touching at $\Gamma$ when $\lambda = 0$; while at the $M_m$ point, there is a gap so long as $\sqrt{\epsilon^2 + 4w_0^2} < \epsilon + 2w_1$, which becomes a triple-band touching at $\sqrt{\epsilon^2 + 4w_0^2} = \epsilon + 2w_1$, and then a quadratic-band touching for $\sqrt{\epsilon^2 + 4w_0^2} > \epsilon + 2w_1$. In this latter region, with both a quadratic-band touching at both $\Gamma$ and $M_m$, turning on finite $\lambda$ produces a trivial band, which requires the two quadratic-band touchings to be of opposite chirality. At finite $\lambda$, where the $\Gamma$ point is gapped, passing from the trivial phase into the spin Hall phase occurs through a Dirac cone touching at $M_m$.

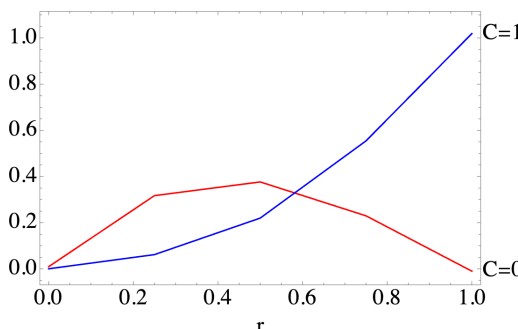

Figure 6: The integrated Berry curvature as a function of the fraction of the total mBZ for the two phases. The bounds of the integrated fraction is defined within a square with side length $rq_S$, where $r \in \{0, \frac{1}{4}, \frac{1}{2}, \frac{3}{4}, 1\}$, centered at $\Gamma$. As $r$ approaches 1, the integrated Berry curvature approaches $C$ [98].

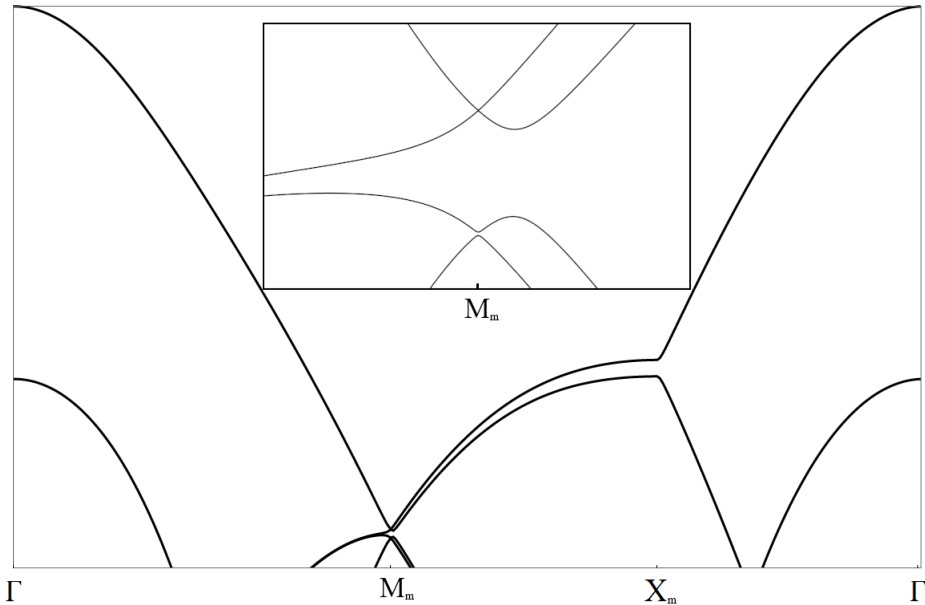

Figure 7: Dirac touching at $M_m$ occurs along the boundary of the spin Hall and trivial phase for finite $\lambda$. The inset shows a zoom on the $M_m$ point.

# 3 Derivations of general continuum theories for the fields at $\Gamma$ and $M$

## 3.1 Irreducible representations at atomic $\Gamma$ and $M$

A complete study of the space group symmetry of Fe-based superconductors was worked out by Cvetkovic and one of us [70]. We follow similarly here, and work within the proper crystallographic representation, which has two Fe's per unit cell. We write the spacing between Fe's of the same unit cell as $a_{\text{FeFe}}$, and the spacing between identical atoms in neighboring unit cells as $l_{\text{Fe}} = \sqrt{2}a_{\text{FeFe}}$. This gives us the lattice vectors $\mathbf{a}_1 = l_{\text{Fe}}(1,0)$ and $\mathbf{a}_2 = l_{\text{Fe}}(0,1)$, which span our Bravais lattice $\mathbf{r} \in \{n\mathbf{a}_1 + m\mathbf{a}_2 | (n,m) \in \mathbb{Z}^2\}$. Additionally, it is necessary to define a half-translation $\mathbf{t} = l_{\text{Fe}}(1/2, 1/2)$, which when combined with an out-of-plane mirror $m_z$ is a (non-symmorphic) crystal symmetry.

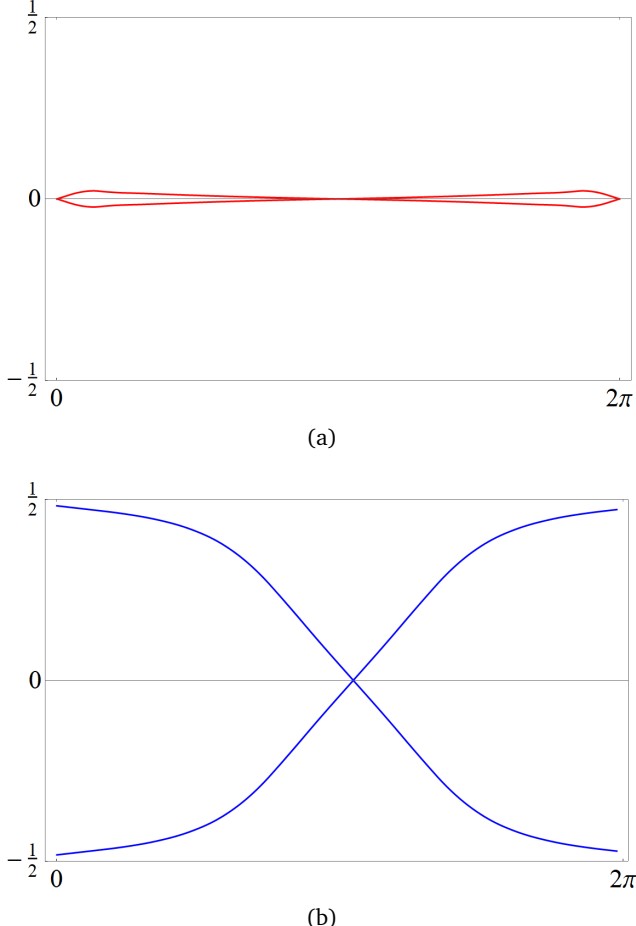

Figure 8: Winding of the charge centers as a function of the mBZ momentum $k_y$ in the (a) trivial and (b) spin Hall phase for both spins.

The reciprocal lattice vectors $\mathbf{Q}_1 = \frac{2\pi}{l_{\text{Fe}}}(1,0)$ and $\mathbf{Q}_2 = \frac{2\pi}{l_{\text{Fe}}}(0,1)$ define our Fe BZ, which has two relevant high symmetry points at $\mathbf{k} = \Gamma$ and $M = \frac{\pi}{l_{\text{Fe}}}(1,1)$. Other BZ's are labelled by $\mathbf{Q} \in \{n\mathbf{Q}_1 + m\mathbf{Q}_2 | (n,m) \in \mathbb{Z}^2\}$. In order to assist the reader, we list here all irreducible representations (irreps) at the $\Gamma$-point and those representations at the $M$-point essential to this paper (i.e. $M_1$ and $M_3$). There are three generators of symmetry, defined to act at an Fe site: two mirrors followed by a fractional translation $tm_X$ and $tm_z$, and one mirror $m_x$. With respect to representations of the group $\mathbf{P}_\Gamma$, it is sufficient to consider all three mirrors without fractional translations: $m_X$, $m_z$, and $m_x$. This is because $\mathbf{P}_\Gamma$ is isomorphic to $\mathbf{D_{4h}}$ [70].

We write the annihilation operator for the fields of an irrep $\mu$ about high symmetry point $\mathcal{Q}$ as $\psi_{\mathcal{Q},\mu,n,\sigma}(\mathbf{k})$; where $\dim(n)$ is equal to the dimension of the irrep (which is $\dim(n) = 2$ for all the irreps in this paper), and $\sigma \in \{\uparrow, \downarrow\}$ labels spin. This is the so-called Kohn-Luttinger (KL) representation of the fields [99]. For the purpose of this paper, we are only interested in spin-orbit coupling which reduces the $SU(2)$ spin symmetry down to $U(1)$, such that the spin decouples into sectors which are necessarily degenerate due to time-reversal. As such, we fix ourselves to a spin-sector (say $\sigma = \uparrow$), and note that the other spin-sector can be acquired via time-reversal $\mathcal{T}$. It is to be thus understood that $\psi_{\mathcal{Q},\mu} = (\psi_{\mathcal{Q},\mu,+}, \psi_{\mathcal{Q},\mu,-})^T$ is an orbital doublet in that fixed spin-sector. The doublet transforms as $\psi_{\mathcal{Q},\mu} \longrightarrow_g \Omega_\mu(g)\psi_{\mathcal{Q},\mu}$, where $\Omega_\mu(g)$ is the unitary representation of generator $g$ acting on irrep $\mu$. At $\Gamma$, $\mu = E_g$; and at $M$, $\mu \in \{M_1, M_3\}$ – which are detailed in Tables 1-2. We write the Pauli matrices acting in the space of the $\dim(n) = 2$ irreps as $\tau_3$, $\tau_1$, and $\tau_2$.

Ultimately, we are interested in defining an analogous operator for the fields in real space, i.e $\psi_{\mathcal{Q},\mu}(\mathbf{r})$. Intuitively, one would expect this to be the Fourier transform of the fields $\psi_{\mathcal{Q},\mu}(\delta\mathbf{k})$ in the limit $|\delta\mathbf{k}| \ll \pi/l_{\text{Fe}}$; thus, even if one does not know the exact form of the wavefunctions at $\mathcal{Q}$, one does know how the fields transform, which is all that is needed to derive a Hamiltonian for the effective field theory of those fields. Nonetheless, we will delve a little deeper here, and show how the effective fields arise within the context of a tight-binding model.

We start by defining the operator which annihilates an electron at an atomic orbital, $d_a(\mathbf{r})$. In the 2-Fe picture, $a$ labels both the type of orbital (here $d$-orbitals) and the sublattice. First expand it in the basis of energy bands $\xi$,

$$d_a(\mathbf{r}) = \int_{\mathbf{k}} \sum_{\xi} e^{i\mathbf{k}\cdot\mathbf{r}} u_{a,\xi}(\mathbf{k}) b_{\xi,\mathbf{k}}, \tag{53}$$

where $\mathbf{k}$ is a momentum in the BZ, and $\int_{\mathbf{k}} \equiv \int \frac{d^2\mathbf{k}}{(2\pi)^2}$. We are interested in an effective field theory for the fields at $\mathcal{Q}$. For the purpose of this demonstration, we introduce a projector $\mathcal{P}_{\mu\Upsilon}$ which projects onto the states within the cutoff $|\delta\mathbf{k}| \leq \Upsilon \ll \pi/l_{\text{Fe}}$, as well as the set of bands $\mathcal{M}_\mu$ containing the irrep $\mu$, i.e $\xi \in \mathcal{M}_\mu$.

$$\mathcal{P}_{\mu\Upsilon} d_a(\mathbf{r}) \mathcal{P}_{\mu\Upsilon} = e^{i\mathcal{Q}\cdot\mathbf{r}} \int_{|\delta\mathbf{k}|\leq\Upsilon} \sum_{\xi \in \mathcal{M}_\mu} e^{i\delta\mathbf{k}\cdot\mathbf{r}} u_{a,\xi}(\mathcal{Q}+\delta\mathbf{k}) b_{\xi,\mathcal{Q}+\delta\mathbf{k}}. \tag{54}$$

Naively, one might try to expand the Bloch states as

$$u_{a,\xi}(\mathcal{Q}+\delta\mathbf{k}) = u_{a,\xi}(\mathcal{Q}) + \delta\mathbf{k}\cdot\boldsymbol{\nabla}_{\mathbf{k}} u_{a,\xi}|_{\mathbf{k}=\mathcal{Q}} + \cdots$$

However, because of the gauge due to translational invariance, the difference $u_{a,\xi}(\mathcal{Q}) - u_{a,\xi}(\mathcal{Q}+\delta\mathbf{k})$ is in general not small, such that the gradient $|\boldsymbol{\nabla}_{\mathbf{k}} u_{a,\xi}|$ is generically large, and therefore $u_{a,\xi}(\mathbf{k})$ is non-analytic. While a resolution may exists in fixing the phase of the Bloch states for a single band, where higher dimensional representations are concerned, the presence of a degeneracy implies a larger gauge freedom at $\mathcal{Q}$.

One can avoid this problem by using the symmetry of the KL fields $\psi_{\mathcal{Q},\mu,n}(\delta\mathbf{k})$ to derive a $\mathbf{k}\cdot\mathbf{p}$-Hamiltonian $h^{\mathcal{Q}}_{n,n'}(\delta\mathbf{k})$, to a desired order in $\delta\mathbf{k}$, which is valid in the vicinity of $\mathcal{Q}$. The weights in the KL representation $u^{\text{KL}}_{a,n}(\mathcal{Q})$ depend on our choice of definition of the fields $\psi_{\mathcal{Q},\mu,n}(\delta\mathbf{k})$, which is arbitrary so long as they transform as the irrep $\mu$. Let unitary $C_{\xi,n}(\delta\mathbf{k})$ diagonalize $h^{\mathcal{Q}}_{n,n'}(\delta\mathbf{k})$. The desired weights $u_{a,\xi}(\delta\mathbf{k})$ are then the linear combination

$$u_{a,\xi}(\delta\mathbf{k}) = \sum_n C_{\xi,n}(\delta\mathbf{k}) u^{\text{KL}}_{a,n}(\mathcal{Q}). \tag{55}$$

Plugging this into Eqn 54 gives us

$$\mathcal{P}_{\mu\Upsilon} d_a(\mathbf{r}) \mathcal{P}_{\mu\Upsilon} = e^{i\mathcal{Q}\cdot\mathbf{r}} \sum_n u^{\text{KL}}_{a,n}(\mathcal{Q}) \int_{\delta\mathbf{k}} e^{i\delta\mathbf{k}\cdot\mathbf{r}} \left( \sum_{\xi \in \mathcal{M}_\mu} C_{\xi,n}(\delta\mathbf{k}) b_{\xi,\mathcal{Q}+\delta\mathbf{k}} \right). \tag{56}$$

The part in parenthesis is the rotation of the Bloch states into the KL representation, which is just our chosen definition of the fields

$$\psi_{\mathcal{Q},\mu,n}(\delta\mathbf{k}) = \sum_{\xi \in \mathcal{M}_\mu} C_{\xi,n}(\delta\mathbf{k}) b_{\xi,\mathcal{Q}+\delta\mathbf{k}}. \tag{57}$$

It can be seen from what remains that the real-space representation of the fields is the Fourier transform of the KL representation,

$$\psi_{\mathcal{Q},\mu,n}(\mathbf{r}) = \int_{\delta\mathbf{k}} e^{i\delta\mathbf{k}\cdot\mathbf{r}} \psi_{\mathcal{Q},\mu,n}(\delta\mathbf{k}). \tag{58}$$

Thus

$$\mathcal{P}_{\mu\Upsilon} d_a(\mathbf{r}) \mathcal{P}_{\mu\Upsilon} = e^{i\mathcal{Q}\cdot\mathbf{r}} \sum_n u_{a,n}^{\mathrm{KL}}(\mathcal{Q}) \psi_{\mathcal{Q},\mu,n}(\mathbf{r}). \tag{59}$$

Since $h_{n,n'}^{\mathcal{Q}}(\delta\mathbf{k})$ is derived to a chosen order in $\delta\mathbf{k}$, it is important to have pre-specified sufficient $\Upsilon \ll \pi/l_{\mathrm{Fe}}$ so that Eqn 55 is valid. But having projected out all non-relevant states, we can get rid of our cutoff by taking $\Upsilon \longrightarrow \infty$, with the caveat that our effective theory is only accurate in the region specified by our original cutoff, i.e $|\delta\mathbf{k}| \ll \pi/l_{\mathrm{Fe}}$.

We may not a-priori know the overlaps of the atomic orbitals with the KL state, $u_{a,n}^{\mathrm{KL}}(\mathcal{Q})$; nor is it necessary to know them to derive an effective theory for the tunneling. Having made clear how the fields transform through Eqn 58, we can proceed with a derivation of the tunneling within the effective theory unhindered. Nonetheless, it may be useful to at least know the ratios between differing $u_{a,n}^{\mathrm{KL}}(\mathcal{Q})$, from which we can deduce the orbital composition of our fields. We do this in the following subsections.

### 3.1.1 $\Gamma$

At $\Gamma$, $e^{i\Gamma\cdot\mathbf{r}} = 1$, thus we only need to consider the generators $m_X$, $m_z$, and $m_x$ acting at an Fe site. Additionally, the only irrep at the $\Gamma$-point transforms identically to an in-plane axial vector (see $E_g$ in Table 1), which is odd under the in-plane mirror $m_z$. Thus we know immediately that $\psi_{\Gamma,n}$ must be composed of $d_{Yz}$ and/or $d_{Xz}$ orbitals. Expanding Eqn 59 for index $a = Yz$ takes the generic form

$$\mathcal{P}_{E_g} d_{Yz}(\mathbf{r}) \mathcal{P}_{E_g} = a \psi_{\Gamma,+}(\mathbf{r}) + b \psi_{\Gamma,-}(\mathbf{r}),$$

with undetermined constants $a, b \in \mathbb{C}$. From transforming under $m_X$, $d_{Yz} \longrightarrow_{m_X} d_{Yz}$ and $\psi_{\Gamma,\pm} \longrightarrow_{m_X} \pm\psi_{\Gamma,\pm}$, we deduce that $b = 0$, which tells us $\psi_{\Gamma,+}$ is $d_{Yz}$. Finally, because $d_{Yz} \longrightarrow_{m_x} d_{Xz}$ and $\psi_{\Gamma,\pm} \longrightarrow_{m_x} -\psi_{\Gamma,\mp}$, we can conclude that at every Fe site

$$\begin{aligned} \psi_{\Gamma,+} &\sim d_{Yz}, \\ \psi_{\Gamma,-} &\sim -d_{Xz}. \end{aligned} \tag{60}$$

### 3.1.2 $M$

The irreducible representations at the corner of the BZ ($\mathbf{P_M}$) are not guaranteed to be isomorphic to a 3D point group if the space group is non-symmorphic [70]. As a consequence, both the action of half-translations, which take an orbital at sublattice $A(B)$ into sublattice $B(A)$, as well as odd integer translations, which accompany a sign change through $e^{iM\cdot\mathbf{a}_1} = e^{iM\cdot\mathbf{a}_2} = -1$, play a key role in the transformation properties of $M_1$ and $M_3$.

We start with the derivation for $M_3$. At this point, we do not yet know which of the 5 $d$-orbitals per 2 sublattices constitute the spinors $\psi_{M_3,\pm}(\mathbf{r})$. However, $\psi_{M_3,\pm}(\mathbf{r})$ is odd under the product $t m_X t m_z$, which preserves sublattice. This tells us that $d_{Xz,A}$ and $d_{Xz,B}$ cannot make up either component of $\psi_{M_3,n}(\mathbf{r})$. Another sublattice-preserving operation, $m_x$, which exchanges $Yz$ and $Xz$, likewise exchanges $\psi_{M_3,+}$ and $\psi_{M_3,-}$ (up to a translation). It then follows that $\psi_{M_3,n}(\mathbf{r})$ can only be orbitals of the $XY$-type. Thus we have

$$\begin{aligned} \mathcal{P}_{M_3} d_{XY,A}(\mathbf{r}) \mathcal{P}_{M_3} &= e^{iM\cdot\mathbf{r}}\big(a'\psi_{M_3,+}(\mathbf{r}) + b'\psi_{m_3,-}(\mathbf{r})\big), \\ \mathcal{P}_{M_3} d_{XY,B}(\mathbf{r}) \mathcal{P}_{M_3} &= e^{iM\cdot\mathbf{r}}\big(c'\psi_{M_3,+}(\mathbf{r}) + d'\psi_{m_3,-}(\mathbf{r})\big), \end{aligned}$$

Table 1: Irreducible Representations of group $\mathbf{P_\Gamma}$ [70].

| $\mathbf{P_\Gamma}$ | $tm_X = \{m_X\vert\frac{1}{2}\frac{1}{2}\}$ | $tm_z = \{m_z\vert\frac{1}{2}\frac{1}{2}\}$ | $m_x = \{m_x\vert 0\}$ |
|---|---|---|---|
| $A_{1g/u}$ | $\pm 1$ | $\pm 1$ | $\pm 1$ |
| $A_{2g/u}$ | $\mp 1$ | $\pm 1$ | $\mp 1$ |
| $B_{1g/u}$ | $\mp 1$ | $\pm 1$ | $\pm 1$ |
| $B_{2g/u}$ | $\pm 1$ | $\pm 1$ | $\mp 1$ |
| $E_{g/u}$ | $\begin{pmatrix} \pm 1 & 0 \\ 0 & \mp 1 \end{pmatrix}$ | $\begin{pmatrix} \mp 1 & 0 \\ 0 & \mp 1 \end{pmatrix}$ | $\begin{pmatrix} 0 & \mp 1 \\ \mp 1 & 0 \end{pmatrix}$ |

with a generic $a', b', c', d' \in \mathbb{C}$. Because $d_{XY,A}(\mathbf{r}) \longrightarrow_{tm_X} d_{XY,B}(\mathbf{r})$ and $d_{XY,B}(\mathbf{r}) \longrightarrow_{tm_X} d_{XY,A}(\mathbf{r} + \mathbf{a}_1 + \mathbf{a}_2)$, in order for the $M_3$-projected orbitals to transform properly under $tm_X$, we need $c' = -a'$ and $d' = b'$.

$$\mathcal{P}_{M_3} d_{XY,A}(\mathbf{r})\mathcal{P}_{M_3} = e^{iM\cdot\mathbf{r}}\big(a'\psi_{M_3,+}(\mathbf{r}) + b'\psi_{m_3,-}(\mathbf{r})\big),$$
$$\mathcal{P}_{M_3} d_{XY,B}(\mathbf{r})\mathcal{P}_{M_3} = e^{iM\cdot\mathbf{r}}\big(-a'\psi_{M_3,+}(\mathbf{r}) + b'\psi_{m_3,-}(\mathbf{r})\big).$$

Lastly, $d_{XY,A}(\mathbf{r}) \longrightarrow_{m_x} d_{XY,A}(\mathbf{r})$ tells us $a' = b'$. This is consistent with its action on the $B$ site, due to the fact that $m_x$ shifts $B$ into itself up to an odd translation, i.e. $d_{XY,B}(\mathbf{r}) \longrightarrow_{m_x} d_{XY,B}(\mathbf{r} - \mathbf{a}_1)$. Thus we are left with

$$\mathcal{P}_{M_3} d_{XY,A}(\mathbf{r})\mathcal{P}_{M_3} = a' e^{iM\cdot\mathbf{r}}\big(\psi_{M_3,+}(\mathbf{r}) + \psi_{M_3,-}(\mathbf{r})\big),$$
$$\mathcal{P}_{M_3} d_{XY,B}(\mathbf{r})\mathcal{P}_{M_3} = a' e^{iM\cdot\mathbf{r}}\big(-\psi_{M_3,+}(\mathbf{r}) + \psi_{M_3,-}(\mathbf{r})\big), \tag{61}$$

or more casually

$$\psi_{M_3,\pm}(\mathbf{r}) \sim e^{-iM\cdot\mathbf{r}}\big(d_{XY,A}(\mathbf{r}) \mp d_{XY,B}(\mathbf{r})\big). \tag{62}$$

We now leave it up to the reader to verify that $M_1$ is composed of $Yz/Xz$ orbitals, and that

$$\mathcal{P}_{M_1} d_{Yz,A}(\mathbf{r})\mathcal{P}_{M_1} = a'' e^{iM\cdot\mathbf{r}}\psi_{M_1,-}(\mathbf{r}),$$
$$\mathcal{P}_{M_1} d_{Yz,B}(\mathbf{r})\mathcal{P}_{M_1} = -a'' e^{iM\cdot\mathbf{r}}\psi_{M_1,-}(\mathbf{r}),$$
$$\mathcal{P}_{M_1} d_{Xz,A}(\mathbf{r})\mathcal{P}_{M_1} = a'' e^{iM\cdot\mathbf{r}}\psi_{M_1,+}(\mathbf{r}),$$
$$\mathcal{P}_{M_1} d_{Xz,B}(\mathbf{r})\mathcal{P}_{M_1} = a'' e^{iM\cdot\mathbf{r}}\psi_{M_1,+}(\mathbf{r}). \tag{63}$$

From which it follows the $M_1$ field transforms like the following composition of orbitals:

$$\psi_{M_1,+}(\mathbf{r}) \sim e^{-iM\cdot\mathbf{r}}\big(d_{Xz,A} + d_{Xz,B}\big), \tag{64}$$
$$\psi_{M_1,-}(\mathbf{r}) \sim e^{-iM\cdot\mathbf{r}}\big(d_{Yz,A} - d_{Yz,B}\big). \tag{65}$$

## 3.2 Continuum limit of iron monolayers with a twist

We derive our models within the effective field theory of the Fe high symmetry points $\Gamma$ and $M$. Let us consider first the theory of a single monolayer experiencing an external superlattice potential (I). Within the framework of the effective field theory at $\mathcal{Q}$, a low-energy field $\psi_{\mathcal{Q},\mu}(\mathbf{r})$ experiences a unique potential $U_{\mathcal{Q},\mu}(\mathbf{r})$, which is a matrix whose dimension is equal to the dimension of the irrep $\mu$. If more than one irrep exists at $\mathcal{Q}$, then additional terms $U_{\mathcal{Q},\mu\mu'}$ which

Table 2: Irreducible Representations $M_1$ and $M_3$ of group $\mathbf{P_M}$ [70].

| $\mathbf{P_M}$ | $tm_X$ | $tm_z$ | $m_x$ |
|---|---|---|---|
| $M_1$ | $\begin{pmatrix} -1 & 0 \\ 0 & -1 \end{pmatrix}$ | $\begin{pmatrix} -1 & 0 \\ 0 & 1 \end{pmatrix}$ | $\begin{pmatrix} 0 & 1 \\ 1 & 0 \end{pmatrix}$ |
| $M_3$ | $\begin{pmatrix} 1 & 0 \\ 0 & -1 \end{pmatrix}$ | $\begin{pmatrix} -1 & 0 \\ 0 & 1 \end{pmatrix}$ | $\begin{pmatrix} 0 & 1 \\ 1 & 0 \end{pmatrix}$ |

account for the scattering between differing irreps due to the potential need be included. The superlattice part of the Hamiltonian therefore takes the general form

$$H_{\mathcal{S},\mathcal{Q}} = \sum_{\mu\mu'} \int d^2\mathbf{r}\, \psi_{\mathcal{Q},\mu}^\dagger(\mathbf{r}) U_{\mu\mu'}(\mathbf{r}) \psi_{\mathcal{Q},\mu'}(\mathbf{r}). \tag{66}$$

In the case of twisted bilayers (II), the Moire potential is generated from the relative displacement of the two planes, which we label $l \in \{1, 2\}$. We work with a representation of the displacement vector $\mathbf{u}_l(\mathbf{x})$, for a monolayer $l$ displaced from a mutual fixed frame (see Fig 9), which is a function of the coordinates $\mathbf{x}$ of that frame. In other words,

$$\mathbf{x} = \mathbf{r}_1 + \mathbf{u}_1 = \mathbf{r}_2 + \mathbf{u}_2. \tag{67}$$

Thus the *relative* displacement can be defined as

$$\mathbf{u}(\mathbf{x}) = \mathbf{u}_1(\mathbf{x}) - \mathbf{u}_2(\mathbf{x}). \tag{68}$$

However, Eqn 68 is only the definition of the relative displacement, and does not tell us how it varies in $\mathbf{x}$. Ultimately we want to consider small-angle rigid rotations, for which the relative displacement takes the form

$$\mathbf{u}(\mathbf{x}) \simeq \theta \hat{\mathbf{z}} \times \mathbf{x}, \qquad \theta \ll 1. \tag{69}$$

But first it will be educational to instead consider cases in which $\mathbf{u}$ is constant, which correspond to non-twisted stacking configurations of the bilayers. For these cases, the tunneling potential between the two layers can be written as a functional of the choice of displacement $\mathbf{u}$ away from the AA stacking configuration (at $\mathbf{u}_{AA} = \mathbf{0}$), i.e $T(\mathbf{u})$. More specifically we write

$$H_{m,\mathcal{Q}} = \sum_{\mu\mu'} \int d^2\mathbf{x}\, \psi_{\mathcal{Q},\mu,1}(\mathbf{x})^\dagger T_{\mu\mu'}(\mathbf{u}) \psi_{\mathcal{Q},\mu',2}(\mathbf{x}) + \text{h.c.}, \tag{70}$$

where $\psi_{\mathcal{Q},\mu,l}(\mathbf{x})$ describes the field of irrep $\mu$ in layer $l$; and $m$ indicates "moire". In order to determine invariant $T(\mathbf{u})$, we treat Eqn 70 as a system of an AA-stacked bilayer in the presence of fluctuating $\mathbf{u}$, such that both $\psi_{\mathcal{Q},\mu,l}$ and necessarily $\mathbf{u}$ (a polar vector) are transformed. Therefore, the action of Eqn 70 under a given mirror operation $g$ follows from:

$$\psi_{\mathcal{Q},\mu,l}(\mathbf{r}_l) \longrightarrow \Omega_\mu(g)\, \psi_{\mathcal{Q},\mu,l}(\mathbf{r}_l'), \tag{71}$$

$$\mathbf{u} \longrightarrow g\mathbf{u}, \tag{72}$$

where $\mathbf{x}' = g\mathbf{x}$, and $\Omega_\mu(g)$ is the unitary spinor representation for the action of $g$ on the irrep $\mu$. (This process is not unlike determining the invariant coupling of a spin vector to a fluctuating

magnetic field $\vec{B} \in \mathbb{R}^3$, i.e $H_{\text{example}} \propto \vec{S} \cdot \vec{B}$, and then setting $\vec{B} = \hat{z}B_0$ to get its coupling to a fixed field.)

Now in the case of small angle twists, Eqn 69, as $\theta \rightarrow 0$ the bilayers approach the AA stacking. This is to say that the effective inter-layer tunneling, which is an analytic function of $\theta$ (i.e an analytic function of the gradients of $\mathbf{u}$) [5], can be expanded in powers of small $\theta$. The leading order term in that expansion is just Eqn 70 with $\mathbf{u}(\mathbf{x})$ equal to Eqn 69. Sub-dominate higher order terms exists, and become more important at larger $\theta$, but we do not include them in this work.

Unlike the case where $\mathbf{u}$ is constant however, there are two important caveats to consider due to the relative rotation of the fields. The first of which is that the action of the fields under the generators of symmetry outlined in Table 1 & 2 are defined for the fields in their respective rotated plane $\psi_{Q,l}(\mathbf{r}_l)$, and are not valid for the fields $\psi_{Q,l}(\mathbf{x})$. In other words, the $d$ orbitals which constitute those fields at a microscopic level are rotated. As discussed in Sec 2.1, any effect from this shows up at sub-leading order $\theta$.

The second effect comes from the fact that the atomic BZ's are relatively rotated between different layers, such that the high symmetry point $Q$ from different layers are located at different momenta. Conveniently, the displacements provides us a map between these two coordinate frames and the fixed lab frame [5]:

$$e^{iQ\cdot\mathbf{x}}\psi_{Q,l}(\mathbf{x}) = e^{iQ\cdot\mathbf{r}_l}\psi_{Q,l}(\mathbf{r}_l),$$
$$\psi_{Q,l}(\mathbf{x}) = e^{-iQ\cdot(\mathbf{x}-\mathbf{r}_l)}\psi_{Q,l}(\mathbf{r}_l),$$
$$\psi_{Q,l}(\mathbf{x}) = e^{-iQ\cdot\mathbf{u}_l}\psi_{Q,l}(\mathbf{r}_l). \tag{73}$$

Because Eqn 73 depends on $\mathbf{u}(\mathbf{x})$, for $|Q| \neq 0$, it contributes an additional phase under $g$ due to $\mathbf{u}(\mathbf{x}) \longrightarrow g\mathbf{u}(\mathbf{x}')$. Thus

$$T_{\mu\mu'}(\mathbf{u}(\mathbf{x}')) = \Omega_\mu(g)^\dagger e^{i(gQ-Q)\cdot\mathbf{u}(\mathbf{x}')}T_{\mu\mu'}(g\mathbf{u}(\mathbf{x}'))\Omega_{\mu'}(g), \tag{74}$$

must hold if we wish for Eqn 70 to be preserved under the mirror operation.

But what about the combination of a mirror followed by a fractional translation? If we return to the definition of our fields in terms of the atomic orbitals Eqn 59, we see that $\mathbf{r}$ corresponds to the location in the Bravais lattice, and does not specify sublattice. The fields are a linear combination of the sublattices, and thus the action of fields under a half-translation $t = \{0|\mathbf{t}\}$ is generally a unitary transformation on the fields, i.e $\psi_{Q,\mu}(\mathbf{r}+\mathbf{t}) = \Omega_\mu(t)\psi_{Q,\mu}(\mathbf{r})$. Since we are considering the crystal planes to be rotated relative to a fixed frame, their fields actually transform as $\psi_{Q,\mu,l}(\mathbf{r}_l + \mathcal{R}^{-1}_{\frac{-2l+3}{2}\theta}\mathbf{t}) = \Omega_\mu(t)\psi_{Q,\mu,l}(\mathbf{r}_l)$. This implies a shift of $\mathbf{x} \longrightarrow \mathbf{x}+\mathbf{t}$, which appears as an order twist-angle contribution from the displacement, i.e $\mathbf{u}(\mathbf{x}+\mathbf{t}) = \mathbf{u}(\mathbf{x}) + \theta\hat{\mathbf{z}} \times \mathbf{t}$. Such a contribution is sub-dominate, being of the order of the twist angle, and therefore does not show up at this order in the derivation. For a combined operation of mirror $g$ followed by fractional translation $t$, it is sufficient to modify Eqn 74 to

$$T_{\mu\mu'}(\mathbf{u}(\mathbf{x}')) = \Omega_\mu(tg)^\dagger e^{i(gQ-Q)\cdot\mathbf{u}(\mathbf{x}')}T_{\mu\mu'}(g\mathbf{u}(\mathbf{x}'))\Omega_{\mu'}(tg). \tag{75}$$

Lastly, we will write $m_{z,l}$ when referring to the action of a mirror in the plane of layer $l$, while leaving the other layer fixed, and without switching the two layers. We then write the combined operation $m_z = m_{z,1} \otimes m_{z,2}$ to imply that the two stacked monolayers are separately but simultaneously inverted about their respective layer planes, without switching the two layers. Separately, we introduce an operation $\mathcal{S}$ which switches the two layers [100], i.e taking $\psi_{Q,\mu,1} \longrightarrow \psi_{Q,\mu,2}$, $\mathbf{u}_1 \longrightarrow \mathbf{u}_2$, and visa-versa. Since the two layers are identical, symmetry demands

$$T^\dagger_{\mu'\mu}(-\mathbf{u}) = T_{\mu\mu'}(\mathbf{u}), \tag{76}$$

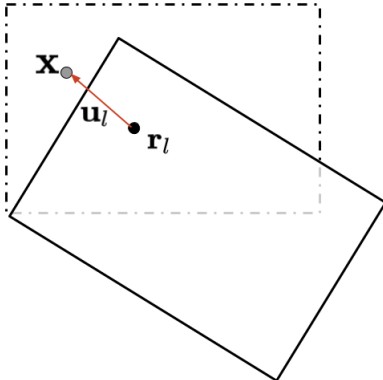

Figure 9: A point in a rotated monolayer **r** and its corresponding point in a fixed (unrotated) coordinate plane $\mathbf{x} = \mathcal{R}_\theta \mathbf{r}$.

which for $\mu = \mu'$ is automatically guaranteed by hermiticity. The layer switching mirror $M_z \equiv m_z \mathcal{S}$ is *not* a symmetry of the AA stacking, unless it is followed by a fractional translation $t M_z$ – the latter which is automatically guaranteed to be a symmetry because $\mathcal{S}$ and $t m_z$ are individually symmetries.

## 3.3 Derivation of monolayer+superlattice at $\Gamma$

The fields at the $\Gamma$-point transform as the two-dimensional irreducible representation $E_g$. Following from Eqn 66, the superlattice potential is in general a $2 \times 2$ matrix,

$$H_{\mathcal{S},\Gamma} = \int d^2\mathbf{x}\, \psi_\Gamma^\dagger(\mathbf{x}) \begin{pmatrix} U_{AA}(\mathbf{x}) & U_{AB}(\mathbf{x}) \\ U_{BA}(\mathbf{x}) & U_{BB}(\mathbf{x}) \end{pmatrix} \psi_\Gamma(\mathbf{x}), \tag{77}$$

which can be more conveniently represented in terms of the Pauli matrices

$$= \int d^2\mathbf{x}\, \psi_\Gamma^\dagger(\mathbf{x}) \big( U_0(\mathbf{x}) + \tau_3 U_3(\mathbf{x}) + \tau_1 U_1(\mathbf{x}) + \tau_2 U_2(\mathbf{x}) \big) \psi_\Gamma(\mathbf{x}). \tag{78}$$

Since the potential is quadratic in fields which transform like $E_g$, each $\tau_\xi$ necessarily transforms as an irreducible representation coming from the product $E_g \otimes E_g$ [70]. More precisely, $\tau_3$, $\tau_1$, and $\tau_2$ transform as $B_{2g}$, $B_{1g}$, and $A_{2g}$ respectively. Technically, there are no necessary symmetry constraints to an external potential applied to a monolayer, simply because we have not yet specified what that potential physically is; however, for the purposes of this paper, we constrain ourselves to those potentials which leave the monolayer effective Hamiltonian invariant. The exception being translational symmetry, which is continuous for the effective monolayer Hamiltonian, and is broken down to periodic by the superlattice. Because the potential is periodic under superlattice shifts, each $U_\xi(\mathbf{x})$ can be Fourier decomposed as

$$\begin{aligned} U_\xi(\mathbf{x}) = {}& \tilde{U}_{\xi,0} + \tilde{U}_{\xi,\mathbf{q}_1} e^{i\mathbf{q}_1 \cdot \mathbf{x}} + \tilde{U}_{\xi,-\mathbf{q}_1} e^{-i\mathbf{q}_1 \cdot \mathbf{x}} + \tilde{U}_{\xi,\mathbf{q}_2} e^{i\mathbf{q}_2 \cdot \mathbf{x}} + \tilde{U}_{\xi,-\mathbf{q}_2} e^{-i\mathbf{q}_2 \cdot \mathbf{x}} \\ & + \tilde{U}_{\xi,(\mathbf{q}_1+\mathbf{q}_2)} e^{i(\mathbf{q}_1+\mathbf{q}_2) \cdot \mathbf{x}} + \tilde{U}_{\xi,-(\mathbf{q}_1+\mathbf{q}_2)} e^{-i(\mathbf{q}_1+\mathbf{q}_2) \cdot \mathbf{x}} \\ & + \tilde{U}_{\xi,(-\mathbf{q}_1+\mathbf{q}_2)} e^{i(-\mathbf{q}_1+\mathbf{q}_2) \cdot \mathbf{x}} + \tilde{U}_{\xi,-(-\mathbf{q}_1+\mathbf{q}_2)} e^{-i(-\mathbf{q}_1+\mathbf{q}_2) \cdot \mathbf{x}} \\ & + \sum_{|\mathbf{q}| \geq 2|\mathbf{q}_1|} \tilde{U}_{\xi,\mathbf{q}} e^{i\mathbf{q} \cdot \mathbf{x}}, \end{aligned} \tag{79}$$

where $\mathbf{q}_1 = q_{\mathcal{S}}(1,0)$ and $\mathbf{q}_2 = q_{\mathcal{S}}(0,1)$ are the superlattice reciprocal lattice vectors, with size $q_{\mathcal{S}} = 2\pi/l_{\mathcal{S}}$; and $\mathbf{q} \in \{ n\mathbf{q}_1 + m\mathbf{q}_2 | (n,m) \in \mathbb{Z}^2 \}$. The first term in the expansion is a

constant corresponding to the spatial average of the potential over the entire superlattice. For $\xi = 0$, this is just a shift in the Fermi energy, while $\xi \neq 0$ terms can be understood to be the constant symmetry breaking terms in the continuum effective field theory. We continue within the assumption that the superlattice potential varies slowly relative to the atomic scale, which means that the Fourier expansion is dominated by its smallest wavevector terms. Thus the largest terms are those with wavevectors which connect adjacent mBZ's (terms 2-5), followed by those which connect next-nearest-neighbor mBZ's along their corners (terms 6-9). We drop all terms with wavevectors satisfying $|\mathbf{q}| \geq 2|\mathbf{q}_1|$.

Since $\mathbf{P}_\Gamma$ is isomorphic to the point group $\mathbf{D_{4h}}$, we need only classify the plane waves by their action under the mirrors (without the fractional translation). The action of a mirror $g$ on the plane wave at $\mathbf{q}$ is to map it into another plane wave at $g\mathbf{q}$, where $|g\mathbf{q}| = |\mathbf{q}|$. Thus the irreducible representations can be classified within a given shell of momenta; and the goal of finding the invariants reduces to finding all products $f_\xi(\mathbf{x})\tau_\xi$ which transform like $A_{1g}$, where we define

$$f_\xi(\mathbf{x}) \equiv \sum_{\mathbf{q}} c_\xi e^{i\mathbf{q}\cdot\mathbf{x}}, \tag{80}$$

for all momentum $\mathbf{q}$ which lie within a given shell (i.e $|\mathbf{q}| = |g\mathbf{q}|$). There are only two such plane-wave representations per shell of momenta, which we classify here:

$$f_{A_{1g}}(\mathbf{x}) = \left[\left(e^{i\mathbf{q}_1\cdot\mathbf{x}} + e^{-i\mathbf{q}_1\cdot\mathbf{x}}\right) + \left(e^{i\mathbf{q}_2\cdot\mathbf{x}} + e^{-i\mathbf{q}_2\cdot\mathbf{x}}\right)\right], \tag{81}$$

$$f_{B_{1g}}(\mathbf{x}) = \left[\left(e^{i\mathbf{q}_1\cdot\mathbf{x}} + e^{-i\mathbf{q}_1\cdot\mathbf{x}}\right) - \left(e^{i\mathbf{q}_2\cdot\mathbf{x}} + e^{-i\mathbf{q}_2\cdot\mathbf{x}}\right)\right], \tag{82}$$

for $|\mathbf{q}| = |\mathbf{q}_1|$, which multiply the identity and $\tau_1$ respectively; and

$$f'_{A_{1g}}(\mathbf{x}) = \left[\left(e^{i(\mathbf{q}_1+\mathbf{q}_2)\cdot\mathbf{x}} + e^{-i(\mathbf{q}_1+\mathbf{q}_2)\cdot\mathbf{x}}\right) + \left(e^{i(-\mathbf{q}_1+\mathbf{q}_2)\cdot\mathbf{x}} + e^{-i(-\mathbf{q}_1+\mathbf{q}_2)\cdot\mathbf{x}}\right)\right], \tag{83}$$

$$f'_{B_{2g}}(\mathbf{x}) = \left[\left(e^{i(\mathbf{q}_1+\mathbf{q}_2)\cdot\mathbf{x}} + e^{-i(\mathbf{q}_1+\mathbf{q}_2)\cdot\mathbf{x}}\right) - \left(e^{i(-\mathbf{q}_1+\mathbf{q}_2)\cdot\mathbf{x}} + e^{-i(-\mathbf{q}_1+\mathbf{q}_2)\cdot\mathbf{x}}\right)\right], \tag{84}$$

for $|\mathbf{q}| = |\mathbf{q}_1 + \mathbf{q}_2|$, which multiply the identity and $\tau_3$. (We introduced a prime – $f_\xi$ and $f'_\xi$ – in order to make clear that the representations are from different shells.) It then follows that there are likewise only two spin-independent invariants per shell of momenta – labelled $w_0$, $w_1$, $w'_0$, and $w'_1$ – which are real numbers due to time-reversal. The symmetry derived potential has the final form:

$$
\begin{aligned}
H_{S,\Gamma}(\mathbf{x}) =\ & w_0 \left[\left(e^{i\mathbf{q}_1\cdot\mathbf{x}} + e^{-i\mathbf{q}_1\cdot\mathbf{x}}\right) + \left(e^{i\mathbf{q}_2\cdot\mathbf{x}} + e^{-i\mathbf{q}_2\cdot\mathbf{x}}\right)\right] \\
& + w_1\ \tau_1\left[\left(e^{i\mathbf{q}_1\cdot\mathbf{x}} + e^{-i\mathbf{q}_1\cdot\mathbf{x}}\right) - \left(e^{i\mathbf{q}_2\cdot\mathbf{x}} + e^{-i\mathbf{q}_2\cdot\mathbf{x}}\right)\right] \\
& + w'_0 \left[\left(e^{i(\mathbf{q}_1+\mathbf{q}_2)\cdot\mathbf{x}} + e^{-i(\mathbf{q}_1+\mathbf{q}_2)\cdot\mathbf{x}}\right) + \left(e^{i(-\mathbf{q}_1+\mathbf{q}_2)\cdot\mathbf{x}} + e^{-i(-\mathbf{q}_1+\mathbf{q}_2)\cdot\mathbf{x}}\right)\right] \\
& + w'_1\ \tau_3\left[\left(e^{i(\mathbf{q}_1+\mathbf{q}_2)\cdot\mathbf{x}} + e^{-i(\mathbf{q}_1+\mathbf{q}_2)\cdot\mathbf{x}}\right) - \left(e^{i(-\mathbf{q}_1+\mathbf{q}_2)\cdot\mathbf{x}} + e^{-i(-\mathbf{q}_1+\mathbf{q}_2)\cdot\mathbf{x}}\right)\right]. \tag{85}
\end{aligned}
$$

Typically, chalcogenide monolayers are grown atop a substrate, such as $SrTiO_3$, which necessarily violates inversion symmetry; however, for FeSe atop $SrTiO_3$, if such inversion breaking exists, it exists at a scale unresolved by the experiments [26]. Nevertheless, if our external superlattice potential arises from a substrate placed atop the chalcogenide monolayer, then it will contribute to the inversion symmetry breaking, and may do so with equal weight to the part of it which is inversion non-violating. Therefore, for the case of a monolayer plus external superlattice, it makes more physical sense to consider all terms which *do not* preserve $tm_z$. However convenient, for the effective field theory at $\Gamma$, no such inversion breaking terms exists at leading order, which follows from the fact that all irreps coming from $E_g \otimes E_g$ are inversion even.

### 3.4 Derivation of monolayer+superlattice at $M$

The derivation for the superlattice potential within the effective field theory at the $M$-point is near-identical to that for $\Gamma$ (see Sec 3.3). Again we consider the fields of a single monolayer, for $M$ which there are two, coupled to a single-particle potential with square lattice period $l_S$. We write these fields $\psi_{M_1}(\mathbf{x})$ and $\psi_{M_3}(\mathbf{x})$, such that the general form for the superlattice potential is

$$
\begin{aligned}
H_{\mathcal{S},M} = &\int d^2\mathbf{x}\, \psi_{M_1}(\mathbf{x})^\dagger \Big( U_{M_1,0}(\mathbf{x}) + \sum_{\mu=1}^3 U_{M_1,\mu}(\mathbf{x})\tau_\mu \Big)\psi_{M_1}(\mathbf{x}) \\
+ &\int d^2\mathbf{x}\, \psi_{M_3}(\mathbf{x})^\dagger \Big( U_{M_3,0}(\mathbf{x}) + \sum_{\mu=1}^3 U_{M_3,\mu}(\mathbf{x})\tau_\mu \Big)\psi_{M_3}(\mathbf{x}) \\
+ &\int d^2\mathbf{x} \sum_\pm \psi_{M_1}^\dagger(\mathbf{x}) U_{E_u,\pm}(\mathbf{x})\big(\tau_0 \pm \tau_3\big)\psi_{M_3}(\mathbf{x}) + \text{h.c.} \\
+ &\int d^2\mathbf{x} \sum_\pm \psi_{M_1}^\dagger(\mathbf{x}) U_{E_g,\pm}(\mathbf{x})\big(\tau_1 \pm i\tau_2\big)\psi_{M_3}(\mathbf{x}) + \text{h.c.}
\end{aligned}
\tag{86}
$$

However, unlike the fields at $\Gamma$, the fields at $M$ do not transform as an irrep of a 3D point group [70, 71], and thus we need necessarily consider the action of fractional translations in our symmetry analysis. Nonetheless, since the superlattice is assumed to be much larger than the Fe-Fe spacing, i.e $a_{\text{FeFe}}/l_S \ll 1$, its action on the plane wave $e^{i\mathbf{q}\cdot\mathbf{x}}$ only produces a small $\mathcal{O}(a_{\text{FeFe}}/l_S)$ phase correction, which can be ignored in this limit.

Notice at this most generic level Eqn 86, we need not only consider the contributions coming from $M_1 \otimes M_1$ and $M_3 \otimes M_3$, which are 1D representations of the space group at $\Gamma$, but additionally two 2D representations of $\mathbf{P}_\Gamma$ coming from $M_1 \otimes M_3 = E_u + E_g$. However, because there are no 2D representations in the plane wave basis described here, and since we are considering only spin-independent terms: $U_{E_g,\pm} = U_{E_u,\pm} = 0$.

Following similarly to Sec 3.3, we consider only terms which preserve the symmetry (except translation) of the effective Hamiltonian for the free fields of the monolayer, and additionally those which violate inversion due to the breaking of $tm_z$. For $M_1 \otimes M_1$ and $M_3 \otimes M_3$, this process is a generalization of that used to derive Eqn 85: (1) classify all possible $f_\xi$ for a given shell of momenta, then (2) find the products $f_\xi\tau_\xi$ which transform like $A_{1g}$ or $A_{2u}$. Since the plane wave representations per shell are the same as in Sec 3.3, we need only classify each $\tau_\xi$ within $M_1 \otimes M_1$ and $M_3 \otimes M_3$ which satisfy criteria (2) for a some $f_\xi(\mathbf{x})$. For $M_1 \otimes M_1$, these are the identity ($A_{1g}$) and inversion-violating $\tau_1$ ($A_{2u}$); and for $M_3 \otimes M_3$, these are likewise the identity ($A_{1g}$) and inversion-breaking $\tau_1$ ($B_{2u}$). One can then check that the potential

$$
\begin{aligned}
H_{\mathcal{S},M} = &\int d^2\mathbf{x}\, \psi_{M_1}^\dagger(\mathbf{x})\, v_{0,M_1}\Big(e^{i\mathbf{q}_1\cdot\mathbf{x}} + e^{-i\mathbf{q}_1\cdot\mathbf{x}} + e^{i\mathbf{q}_2\cdot\mathbf{x}} + e^{-i\mathbf{q}_2\cdot\mathbf{x}}\Big)\psi_{M_1}(\mathbf{x}) \\
+ &\int d^2\mathbf{x}\, \psi_{M_3}^\dagger(\mathbf{x})\, v_{0,M_3}\Big(e^{i\mathbf{q}_1\cdot\mathbf{x}} + e^{-i\mathbf{q}_1\cdot\mathbf{x}} + e^{i\mathbf{q}_2\cdot\mathbf{x}} + e^{-i\mathbf{q}_2\cdot\mathbf{x}}\Big)\psi_{M_3}(\mathbf{x}) \\
+ &\int d^2\mathbf{x}\, \psi_{M_1}^\dagger(\mathbf{x})\, v_{1,M_1}\tau_1\Big(e^{i\mathbf{q}_1\cdot\mathbf{x}} + e^{-i\mathbf{q}_1\cdot\mathbf{x}} - e^{i\mathbf{q}_2\cdot\mathbf{x}} - e^{-i\mathbf{q}_2\cdot\mathbf{x}}\Big)\psi_{M_1}(\mathbf{x}) \\
+ &\int d^2\mathbf{x}\, \psi_{M_3}^\dagger(\mathbf{x})\, v_{1,M_3}\tau_1\Big(e^{i\mathbf{q}_1\cdot\mathbf{x}} + e^{-i\mathbf{q}_1\cdot\mathbf{x}} - e^{i\mathbf{q}_2\cdot\mathbf{x}} - e^{-i\mathbf{q}_2\cdot\mathbf{x}}\Big)\psi_{M_3}(\mathbf{x}),
\end{aligned}
\tag{87}
$$

with real invariants $v_{0,M_1}, v_{0,M_3}, v_{1,M_1}, v_{1,M_3} \in \mathbb{R}$ satisfies our symmetry criteria plus time-reversal.

### 3.5 $\Gamma$-point derivation of moire tunneling for twisted monolayers

When two monolayers are relatively twisted at an arbitrary angle, the resulting moire pattern is generally *not periodic*. However, in the limit of small twist angle $\theta$, one can see by inspection (Fig 1) the emergence of a long-wavelength variation in the pattern with period $l_{\mathcal{S}} \propto \theta^{-1}$. In other words, we expect such a potential is *approximately periodic* under a shift of $\mathbf{x}$ by $\mathbf{R}_j$, i.e $\mathbf{x} \longrightarrow \mathbf{x} + \mathbf{R}_j$. Therefore, in building an effective field theory for slowly varying fields, we might start with the assumption that the effective potential $T(\mathbf{u}(\mathbf{x}))$ is dominated by its periodic part, and thus can be written as a periodic function. Being periodic in $\mathbf{x}$, it would then have the Fourier decomposition

$$T(\mathbf{u}(\mathbf{x})) = \sum_{\mathbf{q}} e^{i\mathbf{q}\cdot\mathbf{x}} \tilde{T}_{\mathbf{q}}, \tag{88}$$

where the superlattice vector $\mathbf{q}_k$ is a moire reciprocal lattice vector satisfying $\mathbf{q}_k \cdot \mathbf{R}_j = 2\pi\delta_{kj}$.

While such a picture of emerging periodicity is intuitive, it falls short in providing a precise connection between the geometry of the microscopic Fe lattices and the emergent moire one. Without which, for instance, we would need to measure $\mathbf{R}_j$ and assume periodicity. However, we need not assume that the moire lattice is periodic in order to guarantee its periodicity, only that the twist angle is sufficiently small as to guarantee the dominate part of the tunneling potential satisfies

$$T(\mathbf{u} + \mathbf{a}_j) = T(\mathbf{u}), \tag{89}$$

for a microscopic monolayer lattice vector $\mathbf{a}_j$. This symmetry Eqn 89 is a statement that the variation due to the displacement is sufficiently small at the atomic scale, that the two layers can be relatively shifted by an atomic unit cell without producing a noticeable change in the inter-layer tunneling. Remembering we label the atomic reciprocal lattice using $\mathbf{Q}$, with basis vectors satisfying $\mathbf{Q}_k \cdot \mathbf{a}_j = 2\pi\delta_{kj}$, then Eqn 89 implies the Fourier expansion

$$T(\mathbf{u}) = \sum_{\mathbf{Q}} e^{i\mathbf{Q}\cdot\mathbf{u}} \tilde{T}_{\mathbf{Q}}. \tag{90}$$

Substituting Eqn 69 into Eqn 90, taking note that $\mathbf{Q}_k \cdot \mathbf{u} = \mathbf{Q}_k \cdot \left(\theta\hat{\mathbf{z}} \times \mathbf{x}\right) = \left(\theta\mathbf{Q}_k \times \hat{\mathbf{z}}\right) \cdot \mathbf{x} = \mathbf{q}_k \cdot \mathbf{x}$, then

$$T(\mathbf{u}(\mathbf{x})) = \sum_{\mathbf{q}} e^{i\mathbf{q}\cdot\mathbf{x}} \tilde{T}_{\mathbf{q}},$$

which is Eqn 88. Thus we have a connection between the microscopic reciprocal lattice vectors and the moire lattice vectors, in terms of the small twist angle [5]:

$$\mathbf{q}_k = \theta\mathbf{Q}_k \times \hat{\mathbf{z}}. \tag{91}$$

We can therefore infer the moire lattice basis vectors through the relation $\mathbf{q}_k \cdot \mathbf{R}_j = 2\pi\delta_{kj}$.

Having now precisely established periodicity through Eqn 91, we can proceed with deriving the tunneling potential (see general form Eqn 70). Specifically, we treat the tunneling as a functional of $\mathbf{u}$, and consider symmetries of the AA-stacked bilayers ($tm_X$, $m_x$, and time-reversal $\mathcal{T}$ are sufficient), but where we additionally transform $\mathbf{u}$. Note that because $\mathcal{Q} = \Gamma = \mathbf{0}$, the dependence on $\mathbf{u}$ coming from the phase in Eqn 73 drops out. This greatly simplifies the form of the tunneling:

$$H_{m,\Gamma}(\mathbf{x}, \mathbf{u}(\mathbf{x})) = \psi_{\Gamma,1}(\mathbf{x})^\dagger T(\mathbf{u}(\mathbf{x}))\psi_{\Gamma,2}(\mathbf{x}) + \text{h.c.} = \psi_{\Gamma,1}(\mathbf{r}_1)^\dagger T(\mathbf{u}(\mathbf{x}))\psi_{\Gamma,2}(\mathbf{r}_2) + \text{h.c.} \tag{92}$$

Consequently, the symmetry analysis follows similarly to Sec 3.3. The tunnelling invariants

have the final form

$$
\begin{aligned}
H_{m,\Gamma} = \int d^2\mathbf{x}\, w_0 \psi_{\Gamma,1}^{\dagger}(\mathbf{x}) \Big[ \big( e^{i\mathbf{q}_1\cdot\mathbf{x}} + e^{-i\mathbf{q}_1\cdot\mathbf{x}} \big) + \big( e^{i\mathbf{q}_2\cdot\mathbf{x}} + e^{-i\mathbf{q}_2\cdot\mathbf{x}} \big) \Big] \psi_{\Gamma,2}(\mathbf{x}) \\
+ w_1 \psi_{\Gamma,1}^{\dagger}(\mathbf{x}) \tau_1 \Big[ \big( e^{i\mathbf{q}_1\cdot\mathbf{x}} + e^{-i\mathbf{q}_1\cdot\mathbf{x}} \big) - \big( e^{i\mathbf{q}_2\cdot\mathbf{x}} + e^{-i\mathbf{q}_2\cdot\mathbf{x}} \big) \Big] \psi_{\Gamma,2}(\mathbf{x}) \\
+ w_0' \psi_{\Gamma,1}^{\dagger}(\mathbf{x}) \Big[ \big( e^{i(\mathbf{q}_1+\mathbf{q}_2)\cdot\mathbf{x}} + e^{-i(\mathbf{q}_1+\mathbf{q}_2)\cdot\mathbf{x}} \big) + \big( e^{i(-\mathbf{q}_1+\mathbf{q}_2)\cdot\mathbf{x}} + e^{-i(-\mathbf{q}_1+\mathbf{q}_2)\cdot\mathbf{x}} \big) \Big] \psi_{\Gamma,2}(\mathbf{x}) \\
+ w_1' \psi_{\Gamma,1}^{\dagger}(\mathbf{x}) \tau_3 \Big[ \big( e^{i(\mathbf{q}_1+\mathbf{q}_2)\cdot\mathbf{x}} + e^{-i(\mathbf{q}_1+\mathbf{q}_2)\cdot\mathbf{x}} \big) - \big( e^{i(-\mathbf{q}_1+\mathbf{q}_2)\cdot\mathbf{x}} + e^{-i(-\mathbf{q}_1+\mathbf{q}_2)\cdot\mathbf{x}} \big) \Big] \psi_{\Gamma,2}(\mathbf{x}) \\
+ \text{h.c.}
\end{aligned}
\tag{93}
$$

Lastly, $w_0$, $w_1$, $w_0'$, and $w_1'$ are real numbers due to $\mathcal{T}$.

Here $w_0$ and $w_1$ are equivalent to the tunnelings of Eqn 3, which we derived from a tight-binding picture in Sec 2.1. In that derivation, only the symmetry of the microscopic atomic $d$ orbitals was used to determine non-zero independent tunnelings, i.e without any explicit reference to the irreducible representations of the chalcogenide space-group symmetries. However, since those irreducible representations have a specific orbital composition dictated by symmetry (see Sec 3.1), any difference between these two derivations is a bit of an illusion. The major difference being that the microscopic approach requires microscopic information, i.e which specific atomic orbitals; whereas the derivation here uses only the fact that the fields at $\Gamma$ transform like $E_g$, without any mention of atomic orbitals.

## 3.6 $M$-point derivation of moire tunelling for twisted monolayers

The precise action of a generator of symmetry on the fields at $M$ depends on whether the field is a function of the fixed ($\psi_{M,l}(\mathbf{x})$) or rotated ($\psi_{M,l}(\mathbf{r}_l)$) coordinate basis. This is because the fields at the $M$-point carry non-zero crystal momentum, and thus $\psi_{M,l}(\mathbf{x}) \neq \psi_{M,l}(\mathbf{r}_l)$, instead following Eqn 73. This makes the derivation of the inter-layer tunneling at the $M$-point most similar to that at the $K$-point in tBG [5]. However, unlike the $K$-point in tBG, which maps into the $(-K)$-point under $\mathcal{T}$, the $M$-point is a time-reversal invariant momentum (TRIM), and therefore maps into itself. Thus in addition to Eqn 74, time-reversal symmetry for $\mathcal{Q} = M$ demands

$$
e^{-2i\mathbf{u}\cdot M} T(\mathbf{u})^* = T(\mathbf{u}).
\tag{94}
$$

We will now proceed with a symmetry derivation of the tunnelling invariants. However, there are two relevant irreps at the $M$ point, $M_1$ and $M_3$, so that we necessarily label our fields $\psi_{M,l,\mu}$ with the additional index $\mu \in \{M_1, M_3\}$. Even though we are working with fields which are irreps of $M$, their self products $M_1 \otimes M_1$ & $M_3 \otimes M_3$ are guaranteed to be 1D irreps of $\Gamma$. Therefore each $U_\mu(\mathbf{u})$ decomposes into a sum over irreps of $\Gamma$:

$$
T_\mu(\mathbf{u}) = T_{\mu,0}(\mathbf{u}) + \sum_{\xi=1}^{3} T_{\mu,\xi}(\mathbf{u}) \tau_\xi.
\tag{95}
$$

Letting $\tau_\xi$ with $\xi = 0$ represent the identity, the Fourier transform is

$$
T_{\mu,\xi}(\mathbf{u}) = \sum_{\mathbf{Q}} e^{i\mathbf{Q}\cdot\mathbf{u}} \tilde{T}_{\mu,\xi,\mathbf{Q}}.
\tag{96}
$$

We choose to label $M = l_{\text{Fe}}^{-1}(\pi, \pi)$, $\mathbf{Q}_1 = l_{\text{Fe}}^{-1}(2\pi, 0)$, and $\mathbf{Q}_2 = l_{\text{Fe}}^{-1}(0, 2\pi)$. However, an important consequence of the non-zero crystal momentum at $M$ is that the tunneling at $\mathbf{Q} = 0$ is

related to that at $\mathbf{Q} \neq 0$ due to phase in Eqn 75, and therefore the various irreducible representations of the plane waves no longer lie within a shell of momentum (as was the case for $\Gamma$). Combining Eqn's 75 and 96 gives the symmetry condition

$$\tau_\xi \sum_\mathbf{Q} e^{i\mathbf{Q}\cdot\mathbf{u}} \tilde{T}_{\mu,\xi,\mathbf{Q}} = \Omega_\mu(tg)^\dagger \tau_\xi \Omega_\mu(tg) \sum_\mathbf{Q} e^{i(gM-M+g\mathbf{Q})\cdot\mathbf{u}} \tilde{T}_{\mu,\xi,\mathbf{Q}}. \tag{97}$$

Because of the orthogonality of the plane waves, each $\tilde{T}_{\mu,\xi,\mathbf{Q}}$ for a given $\mathbf{Q}$ is related to another $\mathbf{Q}' \neq \mathbf{Q}$. This can be thought of as a recursive relation, starting with a given $g\mathbf{Q}$, and shifting it by the change in the crystal momentum due to the generator, i.e $gM - M$. We explicitly list the latter for each $g$ and $\mathcal{T}$ below:

$$m_X\mathbf{M} - \mathbf{M} = -\mathbf{Q}_1 - \mathbf{Q}_2, \tag{98}$$

$$m_z\mathbf{M} - \mathbf{M} = \mathbf{0}, \tag{99}$$

$$m_x\mathbf{M} - \mathbf{M} = -\mathbf{Q}_1, \tag{100}$$

$$\mathcal{T}\mathbf{M} - \mathbf{M} = -\mathbf{Q}_1 - \mathbf{Q}_2. \tag{101}$$

The dominated scattering terms are those which are connected recursively to the $\mathbf{Q} = \mathbf{0}$ term. These correspond to the momenta $\mathbf{0}, -\mathbf{Q}_1, -\mathbf{Q}_2, -\mathbf{Q}_1 - \mathbf{Q}_2$. It is thus sufficient to approximate the tunneling to

$$\tilde{T}_{\mu,\xi}(\mathbf{u}) \simeq \tilde{T}_{\mu,\xi,\mathbf{0}} + \tilde{T}_{\mu,\xi,-\mathbf{Q}_1-\mathbf{Q}_2} e^{i(-\mathbf{Q}_1-\mathbf{Q}_2)\cdot\mathbf{u}} + \tilde{T}_{\mu,\xi,-\mathbf{Q}_1} e^{i(-\mathbf{Q}_1)\cdot\mathbf{u}} + \tilde{T}_{\mu,\xi,-\mathbf{Q}_2} e^{i(-\mathbf{Q}_2)\cdot\mathbf{u}}. \tag{102}$$

Ultimately, there are only two independent invariants per $M_1 \otimes M_1$ and $M_3 \otimes M_3$. We call them $v_0^{(m,M_1)}$, $v_1^{(m,M_1)}$, $v_0^{(m,M_3)}$, and $v_1^{(m,M_3)}$. All other invariants $\tilde{T}_{\xi,\mu,\mathbf{Q}}$ are related to them by symmetry, or else are equal to zero, following

$$\tilde{T}_{\mu,0,\mathbf{0}} = \tilde{T}_{\mu,0,-\mathbf{Q}_1} = \tilde{T}_{\mu,0,-\mathbf{Q}_2} = \tilde{T}_{\mu,0,-\mathbf{Q}_1-\mathbf{Q}_2} \equiv v_0^{(m,\mu)} \in \mathbb{R}, \tag{103}$$

$$\tilde{T}_{\mu,3,\mathbf{0}} = -\tilde{T}_{\mu,3,-\mathbf{Q}_1} = -\tilde{T}_{\mu,3,-\mathbf{Q}_2} = \tilde{T}_{\mu,3,-\mathbf{Q}_1-\mathbf{Q}_2} \equiv v_1^{(m,\mu)} \in \mathbb{R}, \tag{104}$$

$$\text{Else } \tilde{T}_{\mu,\xi,\mathbf{Q}} = 0. \tag{105}$$

In addition to products of the type $M_1 \otimes M_1$ & $M_3 \otimes M_3$, we need to consider tunneling which mixes the representations, i.e per the product $M_1 \otimes M_3$. There are two possible representations which we can form from this product: $E_u$ & $E_g$. Notice that because of the non-trivial transformation of the plane waves due to the phase factor $e^{iM\cdot\mathbf{u}}$, the doublet

$$\left\{ \left(1 - e^{-i(\mathbf{Q}_1+\mathbf{Q}_2)\cdot\mathbf{u}(\mathbf{x})}\right), \left(e^{-i\mathbf{Q}_1\cdot\mathbf{u}(\mathbf{x})} - e^{-i\mathbf{Q}_2\cdot\mathbf{u}(\mathbf{x})}\right) \right\}, \tag{106}$$

transforms like $E_u$, and is *even* under $tm_z$ (which we previously defined to act within a layer and not switch layers – Sec 3.2). By combining this doublet with $E_u$ from $M_1 \otimes M_3$, we can form an additional invariant $v^{(m,13)} \in \mathbb{R}$. However, it is not possible to similarly form an invariant for $E_g$ coming from $M_1 \otimes M_3$. This is because $E_u \otimes E_g$ is odd under $tm_z$.

The symmetry preserving part of the leading-order moire tunneling at $M$ has the complete form

$$
\begin{aligned}
H_{m,M} = & \int d^2\mathbf{x}\,\psi_{M_1,1}^\dagger(\mathbf{x})\Bigg[\left(v_0^{(m,M_1)}+v_1^{(m,M_1)}\tau_3\right)\left(1+e^{-i(\mathbf{Q_1}+\mathbf{Q_2})\cdot\mathbf{u}(\mathbf{x})}\right) \\
& +\left(v_0^{(m,M_1)}-v_1^{(m,M_1)}\tau_3\right)\left(e^{-i\mathbf{Q_1}\cdot\mathbf{u}(\mathbf{x})}+e^{-i\mathbf{Q_2}\cdot\mathbf{u}(\mathbf{x})}\right)\Bigg]\psi_{M_1,2}(\mathbf{x}) \\
& +\int d^2\mathbf{x}\,\psi_{M_3,1}^\dagger(\mathbf{x})\Bigg[\left(v_0^{(m,M_3)}+v_1^{(m,M_3)}\tau_3\right)\left(1+e^{-i(\mathbf{Q_1}+\mathbf{Q_2})\cdot\mathbf{u}(\mathbf{x})}\right) \\
& +\left(v_0^{(m,M_3)}-v_1^{(m,M_3)}\tau_3\right)\left(e^{-i\mathbf{Q_1}\cdot\mathbf{u}(\mathbf{x})}+e^{-i\mathbf{Q_2}\cdot\mathbf{u}(\mathbf{x})}\right)\Bigg]\psi_{M_3,2}(\mathbf{x}) \\
& +i\int d^2\mathbf{x}\,\psi_{M_1,1}^\dagger(\mathbf{x})v^{(m,13)}\Bigg[(1+\tau_3)\left(1-e^{-i(\mathbf{Q_1}+\mathbf{Q_2})\cdot\mathbf{u}(\mathbf{x})}\right) \\
& +(1-\tau_3)\left(e^{-i\mathbf{Q_1}\cdot\mathbf{u}(\mathbf{x})}-e^{-i\mathbf{Q_2}\cdot\mathbf{u}(\mathbf{x})}\right)\Bigg]\psi_{M_3,2}(\mathbf{x}) \\
& -i\int d^2\mathbf{x}\,\psi_{M_3,1}^\dagger(\mathbf{x})v^{(m,13)}\Bigg[(1+\tau_3)\left(1-e^{-i(\mathbf{Q_1}+\mathbf{Q_2})\cdot\mathbf{u}(\mathbf{x})}\right) \\
& +(1-\tau_3)\left(e^{-i\mathbf{Q_1}\cdot\mathbf{u}(\mathbf{x})}-e^{-i\mathbf{Q_2}\cdot\mathbf{u}(\mathbf{x})}\right)\Bigg]\psi_{M_1,2}(\mathbf{x})+\text{h.c.},
\end{aligned}
$$

where the last two lines are related by Eqn 76.

## 3.7 Fitting the moire potential at the $\Gamma$ point

The invariants $w_{0/1,\text{os}}$, $w_{0/1,\text{t}}$, and $t$ not only parametrize a system of twisted bilayers, but in the limit $\theta = 0$, also parametrize the AA-stacking configuration of the bilayer system. More generally, $w_{0/1,\text{os}}$, $w_{0/1,\text{t}}$, and $t$ are the leading order invariants of the effective theory within the more general theory of elasticity for the displacement $\mathbf{u}$ between the two layers. If the gradients of $\mathbf{u}$ – which equal $\theta$ in the case of a rigid twist – are small, then the dominate part of the tunneling is a functional of $\mathbf{u}$ and not its gradients [5, 83], i.e $T(\mathbf{u})$. The same functional $T(\mathbf{u})$ is valid if $\mathbf{u}$ is chosen to be a constant, which physically corresponds to a non-rotated shift away from the AA stacking configuration. The displacement can be chosen to be simple stacking configurations, which produce effects on the spectrum that can be compared with DFT.

In particular, the change in the energies of the bands at $\mathbf{k} = \Gamma$ for three different stacking configurations are needed in order to fit the five invariants: $\mathbf{u}_{\text{AA}} = (0,0)$, $\mathbf{u}_{\text{AB}} = (l_{\text{Fe}}/2, l_{\text{Fe}}/2)$, and $\mathbf{u}_{\frac{1}{2}} = (l_{\text{Fe}}/2, 0)$. To start, let us observe the Hamiltonian for the AA-stacked bilayer system:

$$
H_{\text{AA}} - \epsilon_\Gamma = \begin{pmatrix} 4w_{0,\text{os}} & -i\lambda & t+4w_{0,\text{t}} & 0 \\ i\lambda & 4w_{0,\text{os}} & 0 & t+4w_{0,\text{t}} \\ t+4w_{0,\text{t}} & 0 & 4w_{0,\text{os}} & -i\lambda \\ 0 & t+4w_{0,\text{t}} & i\lambda & 4w_{0,\text{os}} \end{pmatrix} \tag{107}
$$

$$
= 4w_{0,\text{os}} + \lambda\tau_2 + (t+4w_{0,\text{t}})\sigma_1. \tag{108}
$$

The Pauli matrices $\tau_j$ and $\sigma_j$ represent the orbital and layer pseudo-spin sectors respectively. Because no cross terms of the type $\tau_j\sigma_k$ exists which could mix these two different pseudospins, the Hamiltonian can be decoupled into independent subsectors in which $\tau_2$ and $\sigma_1$

are diagonal. This fully diagonalizes the Hamiltonian, which has energies

$$E_{AA,\pm,+\lambda} = \epsilon_\Gamma + 4w_{0,os} + \lambda \pm (t + 4w_{0,t}), \tag{109}$$

$$E_{AA,\pm,-\lambda} = \epsilon_\Gamma + 4w_{0,os} - \lambda \pm (t + 4w_{0,t}). \tag{110}$$

The spectrum for the AB-stacking configuration is identical but with $w_{0,os/t} \rightarrow -w_{0,os/t}$,

$$E_{AB,\pm,+\lambda} = \epsilon_\Gamma - 4w_{0,os} + \lambda \pm (t - 4w_{0,t}), \tag{111}$$

$$E_{AB,\pm,-\lambda} = \epsilon_\Gamma - 4w_{0,os} - \lambda \pm (t - 4w_{0,t}). \tag{112}$$

The on-site potential $w_{0,os}$ is determined as the difference in the average band energy (within a spin-orbit sector) between the two stackings,

$$8w_{0,os} = \left(\frac{E_{AA,+,+\lambda} + E_{AA,-,+\lambda}}{2}\right) - \left(\frac{E_{AB,+,+\lambda} + E_{AB,-,+\lambda}}{2}\right), \tag{113}$$

and the tunneling $w_{0,t}$ sets the variation in the band splitting between the two configurations,

$$8w_{0,t} = \left(\frac{E_{AA,+,+\lambda} - E_{AA,-,+\lambda}}{2}\right) - \left(\frac{E_{AB,+,+\lambda} - E_{AB,-,+\lambda}}{2}\right). \tag{114}$$

The leading-order tunneling $t$ is the average splitting,

$$2t = \left(\frac{E_{AA,+,+\lambda} - E_{AA,-,+\lambda}}{2} + \frac{E_{AB,+,+\lambda} - E_{AB,-,+\lambda}}{2}\right). \tag{115}$$

In order to properly label the bands outputted by DFT, we start with out-of-plane distance between the layers such that the two layers do not see each other. In this limit, the observed bandstructure is that of a monolayer with a layer degeneracy. We then lower the top layer in steps until we reach the out-of-plane distance which minimizes the total DFT energy for the AA stacking.

Having established $t$, we can use the final stacking $\mathbf{u}_{\frac{1}{2}} = (l_{Fe}/2, 0)$ to fit $w_{1,os/t}$. Starting again with the Hamiltonian at $\mathbf{k} = \Gamma$,

$$H_{\frac{1}{2}} - \epsilon_\Gamma = \begin{pmatrix} 0 & -i\lambda + 4w_{1,os} & t & 4w_{1,t} \\ i\lambda + 4w_{1,os} & 0 & 4w_{1,t} & t \\ t & 4w_{1,t} & 0 & -i\lambda + 4w_{1,os} \\ 4w_{1,t} & t & i\lambda + 4w_{1,os} & 0 \end{pmatrix} \tag{116}$$

$$= 4w_{1,os}\tau_1 + \lambda\tau_2 + (t + 4w_{1,t}\tau_1)\sigma_1. \tag{117}$$

Again, we can rotate into a basis which diagonalizes $\sigma_1$,

$$H_{\frac{1}{2},\pm} - \epsilon_\Gamma = 4(w_{1,os} \pm w_{1,t})\tau_1 + \lambda\tau_2 \pm t,$$

which correspond to the layer bonding/anti-bonding sectors discussed in the main text. The eigenvalues are therefore

$$E_{\frac{1}{2},\pm,+} = \epsilon_\Gamma \pm t + 4\sqrt{(w_{1,os} \pm w_{1,t})^2 + (\lambda/4)^2}, \tag{118}$$

$$E_{\frac{1}{2},\pm,-} = \epsilon_\Gamma \pm t - 4\sqrt{(w_{1,os} \pm w_{1,t})^2 + (\lambda/4)^2}. \tag{119}$$

However, due to the small size of $w_{1,os/t}$, and because $w_{1,os/t}$ shows up in quadrature with the much larger $\lambda$, we found it to be impractical to estimate $w_{1,os/t}$ in the presence of spin-orbit. Therefore, we perform the calculation in the absence of spin-orbit (i.e without using fully relativistic pseudopotentials). Setting $\lambda = 0$, we determine the final two invariants as

$$8|w_{1,os} \pm w_{1,t}| = E_{\frac{1}{2},\pm,+} - E_{\frac{1}{2},\pm,-}. \tag{120}$$

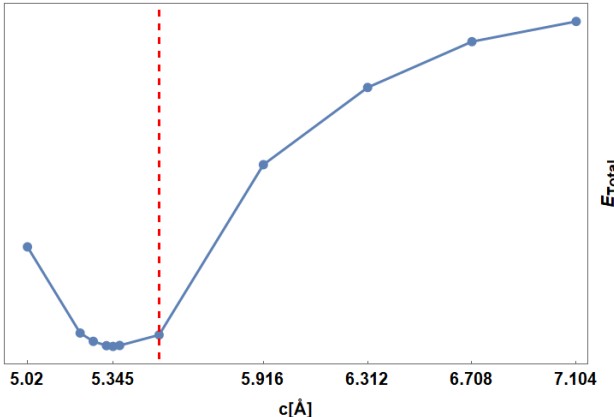

Figure 10: Energy of the AA stacking configuration of bilayer FeSe as a function of inter-layer distance. The red dashed line marks the $c_{\text{bulk}} = 5.5178$ inter-layer distance in the relaxed bulk [33].

## 4 Conclusion

Because the irreducible representations at $\Gamma$ are irreps of $D_{4h}$, the bandstructure reported here for the $\Gamma$ point is generic to other devices constructed from layers with a square lattice geometry [55, 101]. While a quadratic-band touching is a requirement for the spin Hall phase, the Hubbard limit Eqn 8 is still achievable for a single band with a band extremum at $\Gamma$. (This is akin to the $\lambda \to \infty$ limit of a quadratic-band touching.) If such a band minimum/maximum is partially occupied by electrons/holes, such that there is a small Fermi surface at $\Gamma$, and $q_{\mathcal{S}}$ is chosen such that the entire Fermi surface lies in the first superlattice zone, then the physics is that of the $s$-orbital (Fig 2b) Hubbard model.

It is well known that the strong coupling $s$-orbital Hubbard model exhibits antiferromagnetism when the band is half filled. At half filling the dominate Coulomb repulsion can be minimized by placing charges one per site. The tunneling is perturbative, and shows up as a second-order process in which a fermion hops to a neighboring site and back. Mathematically, corrections to the energy coming in at second-order in perturbation theory necessarily bring a minus sign, and therefore lower the energy, selecting out a new ground state from the manifold of one-particle-per-site states. Since such a hopping process is only allowed if neighboring fermions have opposite spin (b/c like-spins are forbidden by Pauli exclusion), the ground state is an antiferromagnet.

However if, for fixed chemical potential, we were to tune $q_{\mathcal{S}}$ such that the Fermi surface lies just outside the first superlattice zone, then the system becomes that of interacting $p$-orbitals [102–104]. This is because Eqn 8 describes a crystal of isotropic quantum harmonic oscillators, whose first excited states are Gaussians multiplied by a form factor $x$ (or $y$). The 2-fold nature of which manifests as the double degeneracy seen as we approach the atomic limit in Fig 4. Thus for fixed chemical potential, decreasing $q_{\mathcal{S}}$ decreases the number of unoccupied states in the $s$-band, until all states are occupied and fermions are forced to occupy the harmonic oscillator's first excited states.

The geometry and nodal structure of these $p$-type single particle states significantly changes the strong-coupling story from the $s$-band case. Here the lobe-like shape of these orbitals makes placing $p_x$ and $p_y$ orbitals on every other site energetically favourable, as it minimizes the anisotropic component of the repulsive Coulomb. Even though the Coulomb dominates at low densities (i.e large $l_{\mathcal{S}}$), the Coulomb anistropy vanishes as neighboring charges look increasingly point-like at large distances. Consequently, the Coulomb anisotropy can be tuned

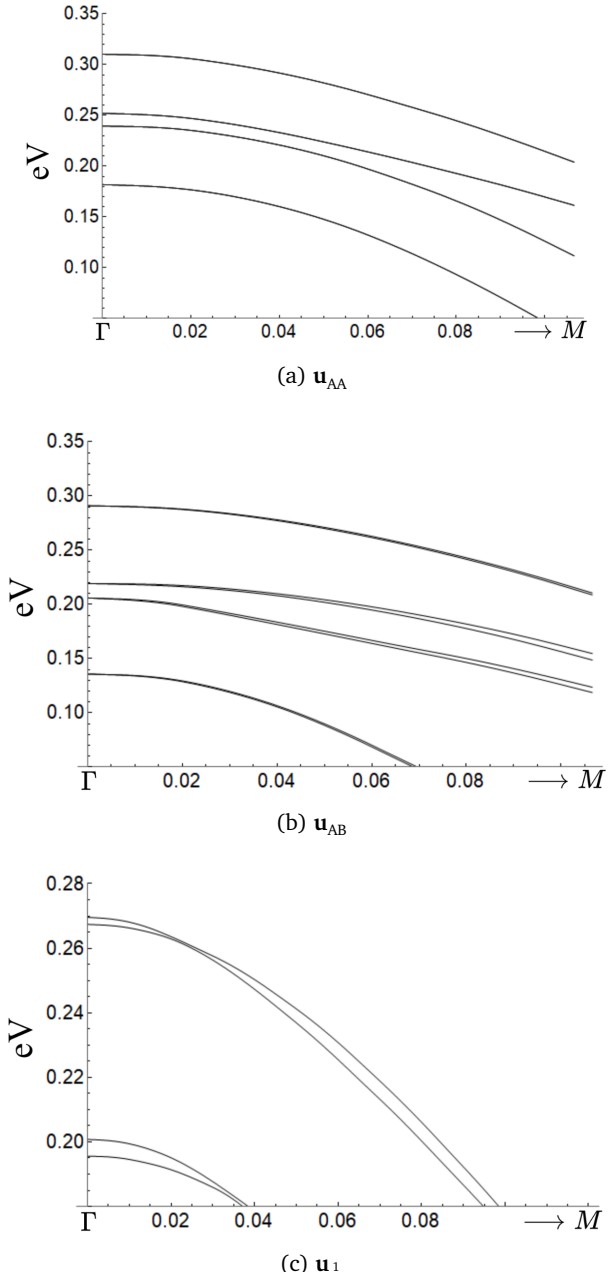

Figure 11: Bands near $\Gamma$ for the (a) AA stacked, (b) AB stacked, and $\mathbf{u}_{\frac{1}{2}} = (\frac{l_{\text{Fe}}}{2}, 0)$ stacking. The momenta are given in units of $\frac{2\pi}{l_{\text{Fe}}}$. The band energies at $\Gamma$ are reported as follows: $\{E_{\text{AA},+,+\lambda}, E_{\text{AA},-,+\lambda}, E_{\text{AA},+,-\lambda}, E_{\text{AA},-,-\lambda}\} = \{310.3, 252.9, 238.5, 181.0\}\text{meV}$, $\{E_{\text{AB},+,+\lambda}, E_{\text{AB},-,+\lambda}, E_{\text{AB},+,-\lambda}, E_{\text{AB},-,-\lambda}\} = \{290.2, 218.4, 204.0, 135.1\}\text{meV}$, and $\left\{E_{\frac{1}{2},+,+}, E_{\frac{1}{2},+,-}, E_{\frac{1}{2},-,+}. E_{\frac{1}{2},-,-}\right\} = \{269.5, 267.5, 200.8, 195.7\}\text{meV}$.

small such that it competes with the second-order tunneling processes. The dominate tunneling process is one in which a fermion hops from $p_x$ ($p_y$) to an adjacent $p_x$ ($p_y$) along the $x$ ($y$) axis. Tunneling of this kind prefers to lower the energy by aligning neighboring $p$ orbitals, in opposition to the Coulomb anisotropy, which prefers them disaligned. It has been argued previously that this competition at second order can give rise to dynamics governed by fourth-order plaquette hopping which supports a gapless bond algebraic liquid phase in bosons [102],

and may be a path toward a tunable non-Fermi liquid state.

In this way described, the Fe-chalcogenides offer a promising pathway toward filling the void left by the success of triangular/hexagonal Hubbard simulators based on TMDs [17–21]. Crucially, this means the potential exploration of exotic phases unique to the square lattice geometry [102]; as well as providing a pathway toward engineering superconductivity driven by repulsive interactions, the existence of which in the 2D Hubbard model has been a point of debate [105]. Recent state-of-the-art density matrix renormalization group (DMRG) and auxilliary field quantum Monte Carlo (AFQMC) found that the square Hubbard model with next-to-nearest-neighbor hopping and repulsive interactions exhibits superconductivity [106, 107]. In our model, the ratio of nearest-neighbor and next-to-nearest neighbor hopping is a tunable function of the twist angle, thus opening up the possibility to directly engineer a quintessential toy model for the cuprates using iron.

This ability for small-angle moire devices to simulate model systems in their ground state is a consequence of the insensitivity of the low-energy fermions to the microscopic intricacies of the moire pattern. While higher-order contributions to the continuum theory for a uniform twist exist – see Sec 3.2, 3.5, and the text surrounding Eqn 22 – they are constrained by geometry to be sub-leading. Such sub-leading terms would have to overcome the energy scale of the leading-order tunneling/potential, in order to close the gap and fundamentally change the states of the band. This is a similar story to the BM model for twisted graphenes [4], which despite neglecting higher order contributions to the continuum theory [5], still succeeds in capturing the magic angle and topology of the bands. Nevertheless, a complete determination of the strong-coupling ground state requires understanding what role kinetic energy plays as a perturbation [83, 84]. Therefore, any future studies of the strong-coupling problem will necessarily need to consider the role of these sub-leading terms in potentially reshaping the stories described above.

Similarly beyond the scope of our continuum theory are the effects due to lattice relaxation and reconstruction. Consider for a moment rigidly rotating a stacked bilayer some $\theta \to 0$ away from its AA stacking, such that $l_S$ is much larger than the sample size. If the AA stacking is stable, then for sufficiently small $\theta$, the stacking is almost everywhere AA, and thus one would expect the bilayer to relax back to AA. If $l_S$ is smaller than the sample size, such that a moire pattern forms with multiple AA, AB, etc stackings locally throughout, then the twist is stable. This is because in order for the bilayer to rotate back into an everywhere AA stacking ($\theta = 0$), it necessarily needs to convert locally AA stacked regions into stacking configurations which are at higher energy. This picture, however, naively considers only rigid deformations of the type described by $\mathbf{u}(\mathbf{x}) = \theta \hat{\mathbf{z}} \times \mathbf{x} + \mathbf{u}_0$. In a real system, $\mathbf{u}(\mathbf{x})$ can adjust at $\mathbf{x}$ in order to locally minimize the competition between the tension within the layers and the energy of the local stacking. So long as the gradients of $\mathbf{u}$ (i.e $\partial_\alpha u_\beta$) remain small, then a continuum expansion remains possible [83, 84], but where $\partial_\alpha u_\beta$ is no longer necessarily constant (as opposed to a rigid twist where it equals $\theta$). However, for small enough $\theta$, there is the possibility that the lattice reconstructs as to grow the AA regions, forming sharp boundaries with large $\partial_\alpha u_\beta$ between AA stacked domains [108].

This phenomena is therefore expected to set a limit on the maximum size of $l_S$ for which a moire continuum theory is still valid. We cannot make any quantitative statements on the size of $l_S^{\max}$ at this time – however, we would like to stress that free-standing iron-based monolayers are unstable. As such, they are necessarily epitaxially grown on a substrate, such as $SrTiO_3$ [96, 97]. The bonding between FeSe and $SrTiO_3$ is not weak, involves a sharing of electrons (doping of FeSe [40]), and is known to strain the iron atoms [40, 95] – leading to a small difference in lattice constant relative to the bulk [40]. Therefore, any relaxation/reconstruction would have to overcome intra-layer tension of the combined FeSe/$SrTiO_3$ layer, making moire devices constructed from them more robust to these effects. This is unlike the case of monolayer TMDs

or graphene, which are not strongly bonded to their substrates (typically hexaboron nitride).

Lastly, while not explicitly studied here, the superlattice potentials/tunnelings derived for the atomic $M$ point pave the way for future theoretical studies. Similar to the $K$ point in graphene, the $M$ point in FeSe is at non-zero crystal momentum. This makes it such that the moire tunneling invariants cannot be expanded as a series of Fourier harmonics $\mathbf{q}$ ordered by increasing $|\mathbf{q}|$. This is because of the fast oscillating plane-wave component of the fields at $M$, $e^{iM\cdot\mathbf{x}}$ in Eqn 1. Since $M$ is rotated in one layer relative to the other, the tunneling between the two layers involves a modulation from the overlap of these waves, i.e $e^{-iM\cdot\mathbf{x}}e^{iR_\theta^{-1}M\cdot\mathbf{x}} = e^{-i(M-R_\theta^{-1}M)\cdot\mathbf{x}}$. Therefore, the plane waves about $M$ do not transform simply under space group symmetries, but instead have a non-trivial behaviour coming from this modulation due to the twist (see Eqn 75). Generically, this gives rise to special points $\mathbf{x}$ corresponding to local stacking configurations $\mathbf{u}(\mathbf{x})$ at which the plane waves in either layer destructively interfere and the tunneling vanishes, dubbed *destructive interference manifolds* [101].

In a previous work [101], studying small-angle twisted square lattices with one orbital per layer, it was argued that the zeros of the tunneling at $M$ act like a potential barrier between moire unit cells. Using a tight-binding calculation, they found that the small-angle limit gives rise to a nested Hubbard model [101] – which can be understood as two decoupled staggered copies of the plain Hubbard limit derived here at $\Gamma$. In fact, remember that our Hubbard limit Eqn 8 is just one copy of a layer-bonded/antibonded band pair. Let us momentarily consider only tunneling ($w_{0,os} = 0$). The $\mathbf{q} = 0$ inter-layer tunneling $t$ sets the energy difference between these band pairs, which are otherwise identical except for experiencing opposite signs of $w_0$. The consequence of this is that they experience a superlattice minima which is relatively shifted between the pairs, and thus are staggered. It is thanks to dominate $t$ that one of these Hubbard copies is pushed down in energy.

Unlike a single-orbital square lattice [101], the fields at $M$ in FeSe are higher-dimensional irreps which *do not* belong to any 3D point group [70,81]. This makes FeSe a potential nursery for novel unexplored moire physics. For example, the underdoped monolayer FeSe hosts Dirac cones directly at the Fermi level [28] (also seen in thin films [109]). These Dirac cones come in pairs along high symmetry lines [110], and are "tilted" owing to a particle-hole breaking component of their dispersion [81]. Dirac cones of this variety have been argued to experience an emergent 2+1 gravity, known as the Painelevé-Gullstrand metric [111–113], such that fermions occupying the cone appear to be in the vicinity of a black hole. A moire superlattice could be used to both tune and isolate these tilted cones into their own bands.

Perhaps most importantly, we found previously that the observed phenomenology of the superconducting state in FeSe can be described within the $\mathbf{k}\cdot\mathbf{p}$ for the $M$ point [81]. It then holds that the invariants derived here can be used to study twisted superconductors [114–122], or the effect of an external superlattice on the superconducting state. This may open a path to robustly determining pairing symmetry [118], which is a question of ongoing debate in FeSe [123]. This was successfully done recently in twisted cuprate thin films [114, 115]. The experimental success in twisted cuprates has inspired efforts to use the $d$-wave nodes to engineer topological superconductivity [121, 122]. Since the Fermi surface in the cuprates extends across the entire BZ, the cuprates are not amenable to a BM-like continuum theory like that we have constructed here. This makes FeSe/SrTiO$_3$ [26] the highest temperature superconductor for which a continuum moire theory can be constructed. Lastly, it remains to be seen if flattening the bands, and therefore increasing the density of states, provides a boost to the already high 65K+ $T_c$ of the monolayer.

## Acknowledgements

We thank Abhay Narayan Pasupathy, Patrick Ledwith, and Kai Sun for helpful comments/discussion. DFT was performed using Quantum ESPRESSO [73].

**Funding information**  P. M. Eugenio and Oskar Vafek are supported by NSF DMR-1916958.

## A  Mathematically flat bands

Surprisingly, even though the twist case (II) decouples into two copies of a monolayer (I), the model still exhibits magic-angle physics. This is best illustrated for a special choice of the $\mathbf{k} \cdot \mathbf{p}$ invariants for the bands of the fields at $\Gamma$. These conditions (elaborated on below), along with taking the limits $\lambda = 0$ and $|w_0/w_1| \ll 1$, produce a Hamiltonian for the moire system which can be written as an anti-holomorphic/holomorphic operator in the off-diagonal. At the magic-angle, the Bloch states for the moire bands at $\mathbf{k} = \Gamma$, which are guaranteed to be zero energy modes by these conditions, generate zeros at both the center and corner of the moire cell. Similar to the chiral limit in graphene [124,125], a zero-energy solution can be found at any $\mathbf{k}$ away from $\Gamma$, thus guaranteeing that the bands are everywhere flat. We explicitly write these solutions in terms of the Jacobi theta functions [126] and the Bloch state at $\Gamma$.

This flattening of the bands occurs independently in both decoupled monolayers, which themselves are independent problems for which the twist angle may be unphysical. Therefore it is better to understand the band flattening as occurring at a *magic cell size $q_S^*$*, rather than a magic angle. We would then describe the flatband limit of case (II) as two copies of (I) with magic cells, the size of which is set by $\theta$ in scenario (II). Within a spin-sector, each copy is composed of two degenerate bands of opposite Chern number – a $\mathbb{Z}_2$ pair. By introducing a finite $\lambda$, the degeneracy is lifted, similar to the action of the sublattice-symmetry breaking in tBG [85], but without violating any symmetries here.

Let us fix ourselves to the following parameters: $a = -2b$, $\mu = 0$, $w_0/w_1 \rightarrow 0$, and $\lambda = 0$. Under these special conditions, Eqn 5 takes the form

$$H_{-,\uparrow}(-i\boldsymbol{\nabla}, \mathbf{x}) = e^{-i\frac{\pi}{4}\tau_1} \begin{pmatrix} 0 & D_{\mathbf{x}} \\ D_{\mathbf{x}}^* & 0 \end{pmatrix} e^{i\frac{\pi}{4}\tau_1}, \tag{A.1}$$

where $D_{\mathbf{x}}$ ($D_{\mathbf{x}}^*$) is an anti-holomorphic (holomorphic) operator in the off-diagonal, defined

$$D_{\mathbf{x}} \equiv -b(\partial_x - i\partial_y)^2 + w_1 U(\mathbf{x}). \tag{A.2}$$

Let us notate the 2-component Bloch states which solves $H_{-,\uparrow}$ as $\Psi_{\mathbf{k}}(\mathbf{x}) = (\phi_{\mathbf{k}}, \chi_{\mathbf{k}})^T$, which satisfy $H_{-,\uparrow}\Psi_{\mathbf{k}}(\mathbf{x}) = E_{\mathbf{k}}\Psi_{\mathbf{k}}(\mathbf{x})$ by definition. In order for there to be perfectly flat bands, then there must exists some $\Psi_{\mathbf{k}}(\mathbf{x})$ s.t $D_{\mathbf{x}}\chi_{\mathbf{k}}(\mathbf{x}) = 0$ and $D_{\mathbf{x}}^*\phi_{\mathbf{k}}(\mathbf{x}) = 0$ holds for every $\mathbf{k}$. By our choice of $\mu$ and $\lambda$ equal to zero, the modes at $\mathbf{k} = \Gamma$ are guaranteed to be zero energy. Therefore, following similarly to Ref [124], if there exists a holomorphic function $f_{\mathbf{k}}(x - iy)$ which is Bloch periodic under superlattice shifts, then it satisfies the equation $D_{\mathbf{x}}f_{\mathbf{k}}(x-iy)\chi_{\Gamma}(x, y) = 0$ with the correct boundary conditions, and is thus our solution.

However, it is well known that functions which are everywhere holomorphic and bounded are necessarily constant. Since a constant function cannot generate the phase required by a Bloch state under periodic shifts, this precludes any $f_{\mathbf{k}}(x - iy)$ which is everywhere holomorphic from being a Bloch state. In order for $f_{\mathbf{k}}(x-iy)$ to satisfy our boundary conditions, it must have poles somewhere in the superlattice unit cell (a.k.a meromorphic) [124]. In general, this prevents us from finding a normalizable solution of the form $\chi_{\mathbf{k}}(\mathbf{x}) = f_{\mathbf{k}}(x - iy)\chi_{\Gamma}(\mathbf{x})$. The

exception occurs when tuned to the magic cell $q_{\mathcal{S}}^*$, where $\chi_\Gamma(\mathbf{x}_0) = 0$ for some $\mathbf{x}_0$ in the unit cell, and the poles from $f_{\mathbf{k}}$ and the zeros from $\chi_\Gamma$ can be made to cancel. Here, depending on the sign of $w_1$, these zeros occur at either the center $\mathbf{x}_0 = 0$ or corner $\mathbf{x}_0 = l_{\mathcal{S}}(\frac{1}{2}, \frac{1}{2})$ of the cell. This corresponds to the shift in the principal Brillouin zone of the hybrid-Wannier states [127], and is analogous to the shift of the Wannier centers for $w_0 \to -w_0$ in the exactly solvable Hubbard limit. In the Hubbard limit, $w_0$ multiplies the cosine potential of Eqn 8, such that flipping its sign flips the potential minima into maxima (and visa versa).

We explicitly construct these meromorphic functions in terms of the Jacobi theta functions $\theta_{a,b}(z|\Omega)$ [126]. Let $\Psi_{-,\mathbf{k}}$ represent the flatband solution to $H_{-,\uparrow}$, which experiences $w_1 > 0$; and similarly $\Psi_{+,\mathbf{k}}$ for $H_{+,\uparrow}$, which experiences $w_1 < 0$. The wavefunctions are

$$\Psi_{\pm,\mathbf{k}}(\mathbf{x}) = \begin{pmatrix} f_\pm(x + iy)\phi_{\pm,\Gamma}(\mathbf{x}) \\ f_\pm(x - iy)\chi_{\pm,\Gamma}(\mathbf{x}) \end{pmatrix}, \tag{A.3}$$

where $f_+$ ($f_-$) has its poles at the center (corner) of the unit cell:

$$f_{+,\mathbf{k}}(z) = \frac{\theta_{\frac{\mathbf{R}_1 \cdot \mathbf{k}}{2\pi} - \frac{1}{2}, -\frac{\mathbf{R}_2 \cdot \mathbf{k}}{2\pi} + \frac{1}{2}}\left(\frac{z}{R_1}|i\right)}{\theta_{-\frac{1}{2}, \frac{1}{2}}\left(\frac{z}{R_1}|i\right)}, \tag{A.4}$$

$$f_{-,\mathbf{k}}(z) = \frac{\theta_{\frac{\mathbf{R}_1 \cdot \mathbf{k}}{2\pi}, -\frac{\mathbf{R}_2 \cdot \mathbf{k}}{2\pi}}\left(\frac{z}{R_1}|i\right)}{\theta_{0,0}\left(\frac{z}{R_1}|i\right)}, \tag{A.5}$$

defining $R_1 \equiv (\mathbf{R}_1)_x + i(\mathbf{R}_1)_y$.

Within a fixed spin sector, the previously neglected spin-orbit $\lambda$ enters the problem in a similar fashion to the sublattice splitting in hexaboron nitride-aligned tBG [85], but without violating any space-group symmetries. Its effect is to split the degenerate flatbands at zero into $C = \pm 1$ pairs of flat bands, separated in energy by $2\lambda$. This is shown in Fig 12.

What then is the roll of finite $\mu$ on the flat bands? Notice that when $\mu = 0$, $h_\Gamma$ satisfies the relation $(\tau_2 h_\Gamma \tau_2)^* = -h_\Gamma$, and therefore exhibits the (single-particle) particle-hole symmetry whose generator is $\mathcal{P} \equiv K\tau_2$; where $K$ represents complex conjugation, and $\tau_2$ is equivalent to a quarter rotation $C_4$. (When $\lambda = 0$, unitary particle-hole symmetry is sufficient, i.e $\{\tau_2, h_\Gamma\} = 0$.) In this limit, the continuum bands of Eqn 2 disperse in opposite directions, unlike the physical system where both bands disperse downward [70]. And the action of $\mathcal{P}$ is to swap the states of the bands.

Introducing $\mu$ has the effect of changing the relative curvature of these bands, breaking the particle-hole symmetry. Nonetheless, $\mu$ being an identity in the orbital basis, the eigenstates of the bands and their action under $\mathcal{P}$ are left unchanged. WLOG, take $b > 0$. For sufficiently negative $\mu$, namely $|\mu|/b \geq 1$, the upper continuum band flips over, such that both bands disperse downward. In flipping over, the upper band necessarily becomes flat, which causes it to be folded infinitely many times onto itself in the mBZ.

Such a singularity obfuscates a direct mathematical connection between the Chern bands of the chiral limit and the Chern bands for $|\mu|/b \geq 1$. We leave it to future research to find a smooth path across the $|\mu|/b \geq 1$ singularity, if one exists, and instead point out that for $\mu = -2830 \text{ meV\AA}^2$ equal to its fitted value for FeSe, we numerically find moire bands which are nearly flat when the cell size is the magic one (see Fig 13a-13b). In this regard, these nearly flat bands for large $\mu$ shadow the perfectly flat bands found at perfect particle-hole symmetry (i.e $\mu = 0$). Therefore, if the value of $\tilde{w}_1$ is greater than what is predicted by DFT, then such a nearly-flat chiral limit shadow may be achievable in FeSe.

Additionally, not only do deviations from $\mu = 0$ ruin the off-diagonal holomorphic/antiholomorphic structure of Eqn A.1 necessary to produce mathematically flat bands, so too do deviations from $a = -2b$. This latter condition amounts to demanding that the bands be

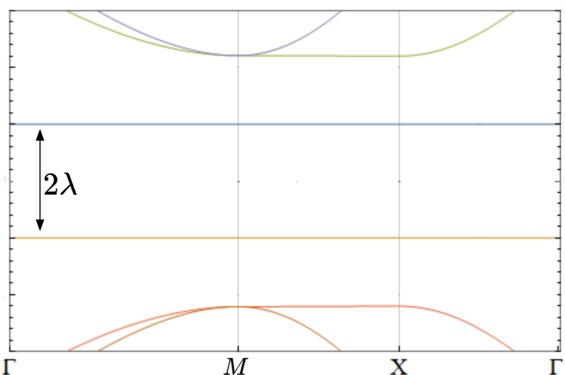

Figure 12: The pair of zero energy flatbands with $(w_0, w_1) = (0, 0.1)$ meV, $(\mu, a, b) = (0, -3440.93, -a/2)$ meVÅ$^2$, $q_S = 0.0148$ Å$^{-1}$, and $\lambda = .1$ meV.

perfectly isotropic about $\Gamma$; however, perfect isotropy is not guaranteed for a quadratic-band touching in a $C_4$ symmetric system. Nevertheless, our fittings for FeSe $a = -2.5b$ are close to this isotropic limit, which is clear from the inset of Fig 1-a, which shows that the energy contours for the upper band are nearly circular. Also, isotropy is guaranteed in the limit of one band, which is to say that increasing $\lambda/(|a|q_S^2)$ suppresses anisotropy, but at the cost of reducing the gap.

Lastly, we would like to point out a separate work proposed recently for engineering similar chiral superlattice bands in a monolayer, but for systems with $C_3$ and $C_6$ symmetry (as opposed to the $C_4$ symmetry here), and where the superlattice is generated from applied periodic strain [128]. Similarly, a quadratic-band touching at $\Gamma$ is found to be required, and the problem is studied for $|\mu|/b \leq 1$, where they found $|\mu|/b = 1$ corresponds to the flat band in the nearest-neighbor-hopping kagome lattice. Unlike the $C_4$ symmetric case, the quadratic-band touchings arising in a $C_3$ symmetric system are guaranteed isotropic; and therefore the bands of Ref [128] do not suffer from the problem described in the previous paragraph. They determined that these flat bands exhibit an ideal quantum geometry for realizing fractional Chern insulators [128], which is not guaranteed for all nearly flat topological bands [125]. It would be interesting to know if similar nearly-flat shadow bands exists in the particle-hole broken regime of Ref [128], and if these nearly-flat shadows meet the mathematical requirements necessary to exhibit topological fractionally-filled phases.

## A.1 Jacobi theta functions, magnetic Bloch states, and their symmetries

A function $f(x + iy)$ which is complex differentiable everywhere in the vicinity of a point is called "holomorphic", or similarly "holomorphic at the point". Let $z = x + iy$. Such functions necessarily satisfy $\frac{df}{dz^*} = 0$ in that point's neighborhood, where $\frac{d}{dz^*} = \frac{1}{2}(\partial_x + i\partial_y)$. This stems from the requirement that the complex derivative defined

$$\frac{df}{dz} \equiv \frac{f(z + dz) - f(z)}{dz} \tag{A.6}$$

is independent of the direction of the infinitesimal step $dz$ in the complex plane – in other words, that the derivatives in the real and imaginary direction do not contradict (are equal). Throughout this manuscript, we refer to both a function $f(x - iy)$ ($f(x + iy)$) and the derivative $\frac{d}{dz^*}$ ($\frac{d}{dz}$) as holomorphic (anti-holomorphic) in order to specify their interrelationship; but describe both $f(x \mp iy)$ as being "holomorphic" in the sense of their property of smoothness.

Non-constant smooth functions which are bounded and periodic over the real numbers, such as $\cos(x)$, are holomorphic yet unbounded in the complex plane. This is clear for

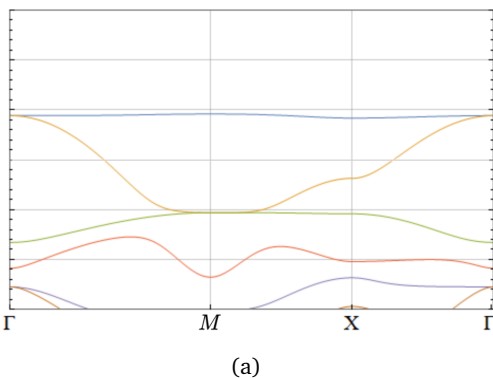
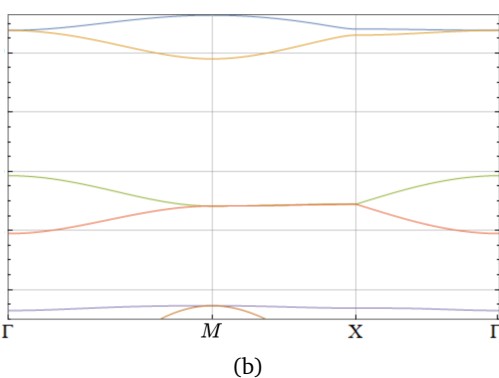

(a)                                        (b)

Figure 13: Nearly flat bands in the region $|\mu|/b \geq 1$. Plotted bands are coloured to distinguish from grey eye-guide lines. (a) The lowest energy superlattice bands with $(w_0, w_1) = (0, 0.1)$ meV, $(\mu, a, b) = (-2830, -3440.93, -a/2)$ meVÅ$^2$, $\lambda = 0$, and $q_{\mathcal{S}} = 0.0148$ Å$^{-1}$. Turning on finite $\lambda$ gaps the blue-orange band touching at $\Gamma$, producing a nearly flat Chern band. (b) For large enough $w_1$, an additional phase exists in which the two lowest energy bands are both gapped from the remote band. Shown plot corresponds to $w_1 = .5$ meV.

$\cos(x + iy) = \cos(x)\cosh(y) - i\sin(x)\sinh(y)$, letting $x, y \in \mathbb{R}$, which diverges in every direction not along the real axis. One can show that the same goes for $\sin(x + iy)$, in fact, this unboundedness is a generic property of functions which are everywhere holomorphic in the complex plane. The only functions which are completely smooth and bounded in $\mathbb{C}$ are constants.

This appears problematic for the construction of holomorphic states with periodic symmetry, however, there is no requirement that periodic functions be smooth. For instance, the ratio of two holomorphic functions such as $1/\cos(x + iy)$, does not blow up at $y = \pm\infty$, but diverges periodically at simple poles corresponding to the zeros of $\cos(x + iy)$. Such functions are referred to as "meromorphic" in literature.

Indeed, there exists a family of functions, the Jacobi theta functions [126], which can be used to construct periodic functions of the type desired. We write these functions as

$$\theta_{a,b}(z|\Omega) = \sum_{n \in \mathbb{Z}} e^{i\pi\Omega(n+a)^2} e^{2\pi i(n+a)(z+b)}, \tag{A.7}$$

where we consider real rational characteristics $a, b \in \mathbb{Q}$, and $\Omega \in \mathbb{C}$. A ratio of these functions was used to construct Eqn A.4, i.e we can define

$$\tilde{\psi}_{a,b}(x, y) \equiv \frac{\theta_{a,b}\left(\frac{x+iy}{R_1}|\Omega\right)}{\theta_{0,0}\left(\frac{x+iy}{R_1}|\Omega\right)}, \tag{A.8}$$

where $\Omega = i\frac{R_2}{R_1}$ guarantees the proper crystal symmetry. The reader can verify that the following symmetries are true for $A, B \in \mathbb{Z}$ [126]:

$$\theta_{a,b}(z + A|\Omega) = e^{2\pi i a A}\theta_{a,b}(z|\Omega), \tag{A.9}$$

$$\theta_{a,b}(z + \Omega B|\Omega) = e^{-2\pi i b B}e^{-i\pi\Omega B^2}e^{-2\pi i B z}\theta_{a,b}(z|\Omega), \tag{A.10}$$

from which it follows $\tilde{\psi}_{a,b}(\mathbf{x} + A\mathbf{R}_1) = e^{2\pi i a A}\tilde{\psi}_{a,b}(\mathbf{x})$ and $\tilde{\psi}_{a,b}(\mathbf{x} + B\mathbf{R}_2) = e^{-2\pi i a B}\tilde{\psi}_{a,b}(\mathbf{x})$. Thus Eqn A.8 transforms as desired for a Bloch state, save for its non-normalizability present in the

periodically-ordered poles in the complex plane, the position of which is set by the rational characteristics of the theta function in the denominator. In the numerator, they represent the crystal momentum as a fraction of the BZ (see Eqn A.4). If normalizability is desired (as it is for any quantum state), another function $\chi(x, y)$ needs to be found with zeros that exactly cancel the simple poles coming from Eqn A.8 when they are multiplied. For our purposes, this occurs for the Bloch state at $\Gamma$, exactly at the magic cell size $q_S^*$. Assuming this condition is met, we can define normalizable $\psi_{a,b} = \tilde{\psi}_{a,b}\chi$.

Unlike trivial Bloch states, the magnetic Bloch states which these functions represent have the property that their phase winds in one of the momentum (here $b$) in a way which depends on the other ($a$): $\psi_{a,b+B}(\mathbf{x}) = e^{-2\pi i a B}\psi_{a,b}(\mathbf{x})$. If we Fourier transform in one direction, we get

$$\omega_{\chi,b}(\mathbf{x}) = \sum_{a\in\mathbb{Z}/N_x\mathbb{Z}} e^{i2\pi a\chi}\psi_{a,b}(x, y), \tag{A.11}$$

where $N_x$ is the number of unit cells in the $x$-direction, $\chi \in \{1, 2, \cdots, N_x\}$, and $\mathbb{Z}/N_x\mathbb{Z} \equiv \{\frac{1}{N_x}, \frac{2}{N_x}, ..., 1\}$. The winding phase therefore guarantees such a function exhibits the behavior that $\omega_{\chi,b+1}(\mathbf{x}) = \omega_{\chi-1,b}(\mathbf{x})$ – a property of the hybrid-Wannier functions with twisted boundary conditions [129]. Such functions are the crystal analog to the continuum Landau levels [85]. The direction of the winding depends on the sign of the Chern number, which can be flipped via a complex conjugation. Therefore, the hybrid-Wannier states constructed from the two flatbands in Fig 12 each have an opposite-spin counterpart that winds in the opposite direction. This in turn means that opposite spins at the edge of the system (or its domains) flow with opposite momentum. This is most easily seen mathematically on a cylinder, where the momentum in one direction is preserved [85, 130].

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
