# Peer review of "Twisted-bilayer FeSe and the Fe-based superlattices"

_SciPost Physics, doi:SciPost Phys. 15, 081 (2023)_

## Round 2 · Referee Report · Anonymous · 2023-3-10

Report
I have reviewed the paper entitled “Twisted-bilayer FeSe and Fe-based superlattices”, by P. Myles Eugenio and Oskar Vafek.
The paper proposes a class of superlattice materials composed of monolayer Fe-chalcogenides and a minimal low-energy model within a K.P effective theory for modes near the Fermi level constructed on the basis of symmetry close to $\Gamma$ and $M$ points.
The authors propose two classes of materials: (I) a single FeSe monolayer experiencing an external periodic potential and (II) uniformly-rotated small-angle twisted bilayers of FeSe, which experience moire inter-layer tunneling. In both cases, a continuum description of the superlattice potential (or tunneling) uses irreducible representations of the space group at $\Gamma$ and $M$.
My overall assessment of the paper is positive, and the presentation is very clear, and previous literature is reported and correctly cited. I would like to give credit to the author for the outstanding contribution that the presented main result makes to the active topic of twisted materials in terms of the contribution it makes to the field. I recommend publication of this work.

---

## Round 2 · Referee Report · Anonymous · 2023-4-10

Report
Before I begin the review, I would like to state that I am not an expert in the specific subject, and therefore do not have a systematic overview of the current literature.
The manuscript introduces superlattice effective models of twisted bilayers in FeSe compound. The introduced models are similar to those that are used for other twisted bilayers/artificial superlattices, but as far as I am aware, the application to FeSe is new. This application is also well-motivated, given the recent experimental and theoretical progress. I expect that the work will have an impact by motivating further experimental studies, and perhaps it would be useful by motivating further theoretical search into Moire materials.
As far as I am aware, the manuscript appears correct, although I certainly recommend that the authors publish the code required to reproduce their numerical results.
As a more significant improvement, I believe the manuscript lacks an overview of the results, a summary of the most important findings, and perhaps a comparative analysis of the Moire FeSe properties versus other materials. Indeed, the manuscript ends on page 28 on a technical discussion rather than a conclusion. I believe that such an overview would greatly benefit the readers in helping them to evaluate the results.
Requested changes
1. Consider publishing the source code for reproducing the manuscript results.
2. Add a summary/review of the results and the comparison with other Moire materials.

---

## Round 2 · Referee Report · Anonymous · 2023-4-10

Strengths
1-Very actual and dynamic subject of the condensed matter of 2D materials
2-The analysis of symmetry properties can be used to derive generic properties of new materials.
3-Results that could lead to experimental realizations.
Weaknesses
1-The article is quite technical, which is not a fault in itself, but it sometimes lacks simpler explanations for a non-specialist reader.
2-The presentation of the manuscript (outline, introduction, conclusion / perspectives) can be improved.
Report
This manuscript proposes a detailed study of the effect of symmetries on low energy flat bands in systems with a superlattice potential (created by an external effect in a monolayer or created by a Moiré in twisted bilayer). The authors use an accurate minimal low-energy models for the bands near the Fermi level that is constructed on the basis of symmetry. Within theses effective theories they address two classes of materials: a single FeSe monolayer experiencing an external periodic potential, and a twisted bilayers of FeSe forming a Moiré pattern. This project is very interesting and it is related to a very actual problem of electronic localization of low energy states by Moiré in 2D materials. The theoretical approach, based on the study of symmetries, is promising, as it should allow to identify generic properties and behaviors that are perhaps characteristic of these new families of materials. The calculations presented seem to me to be correct and the results are a significant advance. So I think this manuscript deserves to be published.
However, I find that the manuscript should be a little more organized to facilitate its reading. After a long introduction that already goes into the details of the study models, the manuscript contains 10 sections and no overall conclusion/perspective section. The manuscript would gain in readability with a shorter and perhaps slightly less technical introduction, a general conclusion/perspective, and a division of the main text in only 2 or 3 sections with subsections. In addition, some parts are quite technical and it would be interesting for readers if some of the main results were highlighted a bit better.
Requested changes
1-As noted in the report, I think the readability of some parts of the manuscript would be greatly improved by reorganizing the text, with fewer sections and subsections. I also think it is important to have a more concise introduction and a general conclusion.
2-It is necessary that authors use the LaTex style of SciPost.
3-Is Section “A Jacobi theta functions, magnetic Bloch states, and their symmetries” part of the Appendix? If so, it should be after the ACKNOWLEDGEMENTS

---

## Round 3 · Referee Report · Anonymous (Referee 1) · 2023-6-6

Report

In that new version of the manuscript, the authors have modified the outline of the article to make it more accessible to the reader. I would have preferred a more concise introduction, but I can understand the authors' arguments. Moreover, they have added a much more detailed conclusion that sets well the article's conclusions in context. As I write in my previous report, this work deserves to be published and I consider the present manuscript ready for publication in SciPost Physics.

---

## Round 3 · Referee Report · Anonymous (Referee 2) · 2023-6-6

Report

In their resubmitted version the authors fully addressed the issues that I brought up, namely the relation to the prior works and a summary discussion. While I still find the summary somewhat demanding to follow, so is the subject. Therefore I recommend the publication of the manuscript as is.

---

## Round 3 · Author Response

We thank both the referees for their time and feedback, and the editor for handling our manuscript. We have updated the formatting and added a conclusion per the referee requests. We agree that the original formatting was unusually, being a consequence of the dual nature of this paper -- one part technical manual, plus an explicit experimental proposal for a given device. Since the latter was arguably more relevant to a larger scientific audience, we focused on it in the lengthy intro, making it as complete as possible.

We do not want to have the reader interested in the proposal bogged down in technical details related to group theory, so to this end, we have kept the lengthy intro in tact; however, we have reorganized the paper into two clear sections, and added a conclusion. We use the conclusion to discuss the relationship to previous works; highlight future work that this manuscript is a foundation for; as well as discuss the limitations of our continuum theory, and what it might mean for future works.

---

## Round 3 · List of Changes

Reorganized sections; added conclusion; moved chiral limit details to appendix (post acknowledgements); and added additional citations relevant to the final discussion.

---

## Editorial Decision

published